# PAC Prediction Sets Under Covariate Shift

**Sangdon Park**
Dept. of Computer & Info. Science
PRECISE Center
University of Pennsylvania
`sangdonp@seas.upenn.edu`

**Edgar Dobriban**
Dept. of Statistics & Data Science
The Wharton School
University of Pennsylvania
`dobriban@wharton.upenn.edu`

**Insup Lee**
Dept. of Computer & Info. Science
PRECISE Center
University of Pennsylvania
`lee@cis.upenn.edu`

**Osbert Bastani**
Dept. of Computer & Info. Science
PRECISE Center
University of Pennsylvania
`obastani@seas.upenn.edu`

## Abstract

An important challenge facing modern machine learning is how to rigorously quantify the uncertainty of model predictions. Conveying uncertainty is especially important when there are changes to the underlying data distribution that might invalidate the predictive model. Yet, most existing uncertainty quantification algorithms break down in the presence of such shifts. We propose a novel approach that addresses this challenge by constructing *probably approximately correct (PAC)* prediction sets in the presence of covariate shift. Our approach focuses on the setting where there is a covariate shift from the source distribution (where we have labeled training examples) to the target distribution (for which we want to quantify uncertainty). Our algorithm assumes given importance weights that encode how the probabilities of the training examples change under the covariate shift. In practice, importance weights typically need to be estimated; thus, we extend our algorithm to the setting where we are given confidence intervals for the importance weights. We demonstrate the effectiveness of our approach on covariate shifts based on DomainNet and ImageNet. Our algorithm satisfies the PAC constraint, and gives prediction sets with the smallest average normalized size among approaches that always satisfy the PAC constraint.

## 1 Introduction

A key challenge in machine learning is quantifying prediction uncertainty. A promising approach is via *prediction sets*, where the model predicts a set of labels instead of a single label. For example, prediction sets can be used by a robot to navigate to avoid regions where the prediction set includes an obstacle, or in healthcare to notify a doctor if the prediction set includes a problematic diagnosis.

Various methods based on these approaches can provide probabilistic correctness guarantees (i.e., that the predicted set contains the true label with high probability) when the training and test distributions are equal—formally, assuming the observations are exchangeable (Vovk et al., 2005; Papadopoulos et al., 2002; Lei et al., 2015) or i.i.d. (Vovk, 2013; Park et al., 2020a; Bates et al., 2021). However, this assumption often fails to hold in practice due to *covariate shift*—i.e., where the input distribution changes but the conditional label distribution remains the same (Sugiyama et al., 2007; Quiñonero-Candela et al., 2009). These shifts can be both natural (e.g., changes in color and lighting) (Hendrycks & Dietterich, 2019) or adversarial (e.g., $\ell_\infty$ attacks) (Szegedy et al., 2014).

We consider unsupervised domain adaptation (Ben-David et al., 2007), where we are given labeled examples from the source domain, but only unlabeled examples from the target (covariate shifted) domain. We propose an algorithm that constructs *probably approximately correct (PAC)* prediction sets under bounded covariate shifts (Wilks, 1941; Valiant, 1984)—i.e., with high probability over

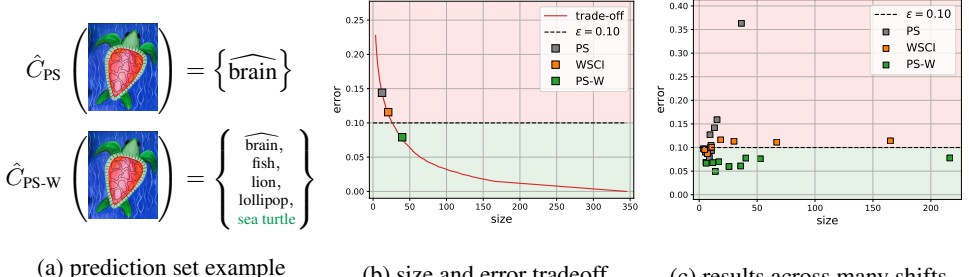

(a) prediction set example      (b) size and error tradeoff      (c) results across many shifts

Figure 1: (a) An example of a covariate shifted image where an existing approach, PS (Park et al., 2020a), constructs prediction sets that do not contain the true label (i.e., "sea turtle"); in contrast, our proposed approach PS-W does. (b) The red curve shows the tradeoff between size and error achieved by different choices of $\tau$ on a single shift; the goal is to be as far to the left as possible without crossing the desired error bound $\varepsilon = 0.1$ shown as the black dashed line. The existing approaches PS and WSCI (Tibshirani et al., 2019) fail to satisfy the desired error bound due to covariate shift, whereas our approach satisfies it. (c) Our approach satisfies the error bound over nine covariate shifts on DomainNet and ImageNet, whereas existing approaches do not.

the training data ("probably"), the prediction set contains the true label for test instances ("approximately correct").

Our algorithm uses importance weights to capture the likelihood of a source example under the target domain. When the importance weights are known, it uses rejection sampling (von Neumann, 1951) to construct the prediction sets. Often, the importance weights must be estimated, in which case we need to account for estimation error. When only given confidence intervals around the importance weights, our algorithm constructs prediction sets that are robust to this uncertainty.

We evaluate our approach in two settings. First, we consider *rate shift*, where the model is trained on a broad domain, but deployed on a narrow domain—e.g., an autonomous car may be trained on both daytime and nighttime images but operate at night. Even if the model continues to perform well, this covariate shift may invalidate the prediction sets. We show that our approach constructs valid prediction sets under rate shifts constructed from DomainNet (Peng et al., 2019), whereas several existing approaches do not. Second, we consider *support shift*, where the model is trained on one domain, but evaluated on another domain with shifted support. We train ResNet101 (He et al., 2016) using unsupervised domain adaptation (Ganin et al., 2016) on both ImageNet-C synthetic perturbations (Hendrycks & Dietterich, 2019) and PGD adversarial perturbations (Madry et al., 2017) to ImageNet (Russakovsky et al., 2015). Our PS-W algorithm satisfies the PAC constraint, and gives prediction sets with the smallest average normalized size among approaches that always satisfy the PAC constraint. See Figure 1 for a summary of our approach and results.

**Related work.** Our work is related to conformal prediction (Vovk et al., 2005; Balasubramanian et al., 2014) without a shift, where the goal is to construct models that predict sets of labels designed to include the true label with high probability (instead of predicting individual labels) under the assumption that the source and target distributions are the same. In particular, our setting is related to *inductive* (or *split*) conformal prediction (Papadopoulos et al., 2002; Vovk, 2013; Lei et al., 2015), where the training set is split into a *proper training set* used to train a traditional predictive model, and a *calibration set* used to construct the prediction sets. The general approach is to train a model $f : \mathcal{X} \times \mathcal{Y} \to \mathbb{R}$ that outputs *conformity scores*, and then to choose a threshold $\tau \in \mathbb{R}$ that satisfies a correctness guarantee, where the corresponding prediction sets are $C(x) = \{y \in \mathcal{Y} \mid f(x, y) \geq \tau\}$ (Park et al., 2020a; Gupta et al., 2021); this notation is defined in Section 2.

Several kinds of correctness guarantees under no shift have been considered. One possibility is *input conditional* validity (Vovk, 2013; Barber et al., 2020), which ensures correctness for *all* future covariates $x$, with high probability only over the conditional label distribution $p(y \mid x)$. This guarantee is very strong, and hard to ensure in practice. A weaker notion is *approximate* (Lei & Wasserman, 2014; Barber et al., 2020) or *group* (Romano et al., 2019) conditional validity, which ensures correctness with high probability over $p(y \mid x)$ as well as some distribution $p(x)$ over a subgroup. Finally, *unconditional validity* ensures only correctness over the joint distribution $p(x, y)$. We focus on unconditional validity, though our approach extends readily to group conditional validity.

A separate issue, arising in the no shift setting, is how to condition on the calibration set $Z$. A conventional goal is unconditional validity, over the distribution $p(x, y, Z)$. We refer to this strategy as *fully unconditional validity*. However, the guarantee we consider uses a separate confidence level for the training data, which is called a *training conditional guarantee* (Vovk, 2013); this correctness notion is equivalent to a PAC correctness guarantee (Park et al., 2020a), and is also equivalent to the standard "content" guarantee with a certain confidence level for tolerance regions (Wilks, 1941; Fraser, 1956). We build on Park et al. (2020a), which formulates the problem of choosing $\tau$ as learning a binary classifier where the input and parameter spaces are both one-dimensional; thus, the correctness guarantee corresponds to a PAC generalization bound. This approach is widely applicable since it can work with a variety of objectives (Bates et al., 2021).

Recent work has extended inductive conformal prediction to a setting with covariate shift (Tibshirani et al., 2019; Lei & Candès, 2020); similarly, Podkopaev & Ramdas (2021) considers conformal prediction under label shift (Lipton et al., 2018), i.e., assuming the conditional probabilities $p(x \mid y)$ do not change. These approaches start from the assumption that the true importance weights (IWs) are known (Tibshirani et al., 2019; Podkopaev & Ramdas, 2021), or assume some properties of the estimated IWs (Lei & Candès, 2020), whereas our approach considers a special form of "ambiguity" in the estimated IWs. Furthermore, they are focused on fully unconditional validity, whereas we obtain PAC prediction sets. In addition, Cauchois et al. (2020) designs confidence sets that are robust to *all* distribution shifts with bounded $f$-divergence; in contrast, we consider the unsupervised learning setting where we have unlabeled examples from the target distribution. We provide additional related work in Appendix A.

## 2 BACKGROUND ON PAC PREDICTION SETS

We give background on PAC prediction sets (Park et al., 2020a); we also re-interpret this approach using Clopper-Pearson confidence intervals (Clopper & Pearson, 1934), which aids our analysis.

### 2.1 PAC PREDICTION SETS ALGORITHM

Let $x \in \mathcal{X}$ be covariates and $y \in \mathcal{Y}$ be labels. We consider a source distribution $P$ over $\mathcal{X} \times \mathcal{Y}$ with probability density function (PDF) $p(x, y)$.[1] A *prediction set*[2] is a set-valued function $C : \mathcal{X} \to 2^{\mathcal{Y}}$.

**Inputs.** We are given a held-out calibration set $S_m$ of i.i.d. samples $(x_i, y_i) \sim P$ for $i \in [m] \coloneqq \{1, \ldots, m\}$, written as $S_m \sim P^m$, and a score function $f : \mathcal{X} \times \mathcal{Y} \to \mathbb{R}_{\geq 0}$. For example, $f(x, y)$ can be a prediction for the probability that $y$ is the label for $x$. The score function can be arbitrary, but score functions assigning higher scores to likely labels yield smaller prediction sets.

**PAC prediction set.** A *PAC prediction set* is a set-valued function $C : \mathcal{X} \to 2^{\mathcal{Y}}$ satisfying the following two conditions. First, $C$ is *approximately correct* in the sense that its expected error (failing to contain the true label) is bounded by a given $\varepsilon \in (0, 1)$, i.e.,

$$L_P(C) \coloneqq \mathbb{E}_{(x,y)\sim P} \left[ \mathbb{1}(y \notin C(x)) \right] = \mathbb{P}_{(x,y)\sim P}[y \notin C(x)] \leq \varepsilon.$$

Second, noting that $C_{S_m}$ is constructed based on a calibration set $S_m \sim P^m$, the condition that $C_{S_m}$ is approximately correct must be satisfied with high probability—i.e., for given $\delta \in (0, 1)$, we have

$$\mathbb{P}_{S_m \sim P^m} \left[ L_P(C_{S_m}) \leq \varepsilon \right] \geq 1 - \delta.$$

Our goal is to devise an algorithm for constructing a PAC prediction set $C$. Letting $C(x) = \mathcal{Y}$ for all $x \in \mathcal{X}$ always satisfies both conditions above, but this extreme case would be uninformative if used as a prediction set. Therefore, we additionally want to minimize the expected size $\mathbb{E}[S(C(x))]$ of the prediction sets $C(x)$, where $S : 2^{\mathcal{Y}} \to \mathbb{R}_{\geq 0}$ is a size measure, which is application specific (e.g., the cardinality of a set in classification); however, we only rely on the monotonicity of the size measure with respect to the prediction set parameterization in construction.

---

[1]All quantities that we define are measurable with respect to a fixed $\sigma$-algebra on $\mathcal{X} \times \mathcal{Y}$; for instance, $p$ is the density induced by a fixed $\sigma$-finite measure. To be precise, we consider a probability measure $P$ defined with respect to the base measure $M$ on $\mathcal{X} \times \mathcal{Y}$; then, $p = \mathrm{d}P/\mathrm{d}M$ is the Radon-Nykodym derivative of $P$ with respect to $M$. For classification, $M$ is the product of a Lebesgue measure on $\mathcal{X}$ and a counting measure on $\mathcal{Y}$.

[2]We use the term "prediction set" to denote both the set-valued function and a set output by this function.

**Algorithm.** To construct $C$, we first define the search space of possible prediction sets along with the size measure $S$. We parameterize $C$ by a scalar $\tau \in \mathcal{T} \coloneqq \mathbb{R}_{\geq 0}$ as

$$C_\tau(x) = \{y \in \mathcal{Y} \mid f(x, y) \geq \tau\},$$

i.e., $\tau$ is the threshold on $f(x, y)$ above which we include $y$ in $C(x)$. Importantly, $\tau$ controls the tradeoff between size and expected error. The reason is that if $\tau_1 \leq \tau_2$, then $C_{\tau_2}(x) \subseteq C_{\tau_1}(x)$ for all $x \in \mathcal{X}$. Thus, size is monotonically *decreasing* in $\tau$—i.e., $S(C_{\tau_2}(x)) \leq S(C_{\tau_1}(x))$ for all $x \in \mathcal{X}$, and error is monotonically *increasing* in $\tau$—i.e., $L_P(C_{\tau_1}) \leq L_P(C_{\tau_2})$. See Figure 1b for an illustration, and Park et al. (2020a) and Gupta et al. (2021) for details.

As a consequence, a typical goal is to construct $C_\tau$ that provably contains the true label with high probability, while empirically minimizing size (Vovk et al., 2005; Gupta et al., 2021). In our setting, we want to maximize $\tau$ (equivalently, minimize expected size) subject to the constraint that $C_\tau$ is PAC. Let $\bar{L}_{S_m}(C_\tau) \coloneqq \sum_{(x,y) \in S_m} \mathbb{1}(y \notin C_\tau(x))$ be the empirical error count on $S_m$, and $F(k; m, \varepsilon) = \sum_{i=0}^{k} \binom{m}{k} \varepsilon^i (1 - \varepsilon)^{m-i}$ be the cumulative distribution function (CDF) of the binomial distribution $\text{Binom}(m, \varepsilon)$ with $m$ trials and success probability $\varepsilon$. In prior work, Park et al. (2020a) constructs $C_{\hat{\tau}}$ by solving the following problem:

$$\hat{\tau} = \max_{\tau \in \mathcal{T}} \tau \quad \text{subj. to} \quad \bar{L}_{S_m}(C_\tau) \leq k(m, \varepsilon, \delta), \tag{1}$$

where $k(m, \varepsilon, \delta) = \max_{k \in \mathbb{N} \cup \{0\}} k$ subj. to $F(k; m, \varepsilon) \leq \delta$. Intuitively, the PAC guarantee via this construction is related to the Binomial distribution; $\bar{L}_{S_m}(C)$ has distribution $\text{Binom}(m, L_P(C))$ (given a fixed $C$), since $\mathbb{1}(y \notin C(x))$ has a $\text{Bernoulli}(L_P(C))$ distribution when $(x, y) \sim P$. Thus, $k(m, \varepsilon, \delta)$ defines a "confidence interval" such that if $\bar{L}_{S_m}(C) \leq k(m, \varepsilon, \delta)$, then $L_P(C) \leq \varepsilon$ with probability at least $1 - \delta$. Below, we formalize this intuition by drawing a connection to the Clopper-Pearson confidence interval.

## 2.2 Clopper-Pearson Interpretation

We interpret (1) using the Clopper-Pearson (CP) upper bound $\overline{\theta}(k; m, \delta) \in [0, 1]$ (Clopper & Pearson, 1934; Park et al., 2021). This is an upper bound on the true success probability $\mu$, constructed from a sample $k \sim \text{Binom}(m, \mu)$, which holds with probability at least $1 - \delta$, i.e., $\mathbb{P}_{k \sim \text{Binom}(m, \mu)}[\mu \leq \overline{\theta}(k; m, \delta)] \geq 1 - \delta$, where

$$\overline{\theta}(k; m, \delta) \coloneqq \inf \{\theta \in [0, 1] \mid F(k; m, \theta) \leq \delta\} \cup \{1\}. \tag{2}$$

Intuitively, for a fixed $C$, $\bar{L}_{S_m}(C) \sim \text{Binom}(m, L_P(C))$, so the true error $L_P(C)$ is contained in the CP upper bound $\overline{\theta}(\bar{L}_{S_m}(C); m; \delta)$ with probability at least $1 - \delta$. Intuitively, we can therefore choose $\tau$ so that this upper bound is $\leq \varepsilon$. However, we need to account for generalization error of our estimator. To this end, we have the following (see Appendix D.1 for a proof and Algorithm 2 in Appendix E for implementation details on the corresponding algorithm):

**Theorem 1** *Let $U_{CP}(C, S_m, \delta) \coloneqq \overline{\theta}(\bar{L}_{S_m}(C); m; \delta)$, where $\overline{\theta}$ is defined in (2). Let $\hat{\tau}$ be the solution of the following problem*[3]:

$$\hat{\tau} = \max_{\tau \in \mathcal{T}} \tau \quad \text{subj. to} \quad U_{CP}(C_\tau, S_m, \delta) \leq \varepsilon. \tag{3}$$

*Then, we have $\mathbb{P}_{S_m \sim P^m}[L_P(C_\tau) \leq \varepsilon] \geq 1 - \delta$ for any $\tau \leq \hat{\tau}$.*

# 3 PAC Prediction Sets Under Covariate Shift

We extend the PAC prediction set algorithm described in Section 2 to the covariate shift setting. Our novel approach combines rejection sampling (von Neumann, 1951) with Theorem 1.

---

[3]We consider a trivial solution $\tau = 0$ when (3) is not feasible, but omitting here to avoid clutter.

### 3.1 PROBLEM FORMULATION

We assume labeled training examples from the *source distribution $P$* are given, but want to construct prediction sets that satisfy the PAC property with respect to a (possibly) shifted *target distribution $Q$*. Both of these are distributions over $\mathcal{X} \times \mathcal{Y}$; denote their PDFs by $p(x, y)$ and $q(x, y)$, respectively. The challenge is that we are only given unlabeled examples from $Q$.

**Preliminaries and assumptions.** We denote the likelihood ratio of covariate distributions by $w^*(x) := q(x)/p(x)$, also called the *importance weight (IW)* of $x$. Our main assumption is the *covariate shift* condition, which says the label distributions are equal (i.e., $p(y \mid x) = q(y \mid x)$), while the covariate distributions may differ (i.e., we can have $p(x) \neq q(x)$).

**Inputs.** We assume a labeled calibration set $S_m$ consisting of i.i.d. samples $(x_i, y_i) \sim P$ (for $i \in [m]$) is given, and a score function $f : \mathcal{X} \times \mathcal{Y} \to \mathbb{R}_{\geq 0}$. For now, we also assume we have the true importance weights $w_i^* := w^*(x_i)$ for each $(x_i, y_i) \sim P$, as well as an upper bound $b \geq \max_{x \in \mathcal{X}} w^*(x)$ on the IWs. In Sections 3.3 and Appendix B, we describe how to estimate these quantities in a way that provides guarantees under smoothness assumptions on the distributions.[4]

**Problem.** Our goal is to construct $C_{S_m}$ that is a PAC prediction set under $Q$—i.e.,

$$\mathbb{P}_{S_m \sim P^m} \left[ L_Q(C_{S_m}) \leq \varepsilon \right] \geq 1 - \delta,$$

where $L_Q(C) := \mathbb{P}_{(x,y) \sim Q}[y \notin C(x)]$ is the error of $C$ on $Q$, while empirically controlling its size.

### 3.2 REJECTION SAMPLING STRATEGY

Our strategy is to use rejection sampling to convert $S_m$ consisting of i.i.d. samples from $P$ into a labeled calibration set consisting of i.i.d. samples from $Q$.

**Rejection sampling.** Rejection sampling (von Neumann, 1951; Owen, 2013; Rubinstein & Kroese, 2016) is a technique for generating samples from a target distribution $q(x)$ based on samples from a proposal distribution $p(x)$. Given importance weights (IWs) $w^*(x)$ and an upper bound $b \geq \max_{x \in \mathcal{X}} w^*(x)$, it constructs a set of i.i.d. samples from $q(x)$. Typically, rejection sampling is used when the proposal distribution is easy to sample compared to the target distribution. In contrast, we use it to convert samples from the source distribution into samples from the target distribution.

In particular, our algorithm takes the proposal distribution to be the source covariate distribution $P_X$, and the target distribution to be our target covariate distribution $Q_X$. Let $V_i \sim U := \text{Uniform}([0, 1])$ be i.i.d., $V := (V_1, \ldots, V_m)$, and $\vec{w}^* := (w_1^*, \ldots, w_m^*)$ with $w_i^* = w^*(x_i)$ as defined before. Then, given $S_m$, it uses rejection sampling to construct a randomly chosen set of $N$ samples

$$T_N(S_m, V, \vec{w}^*, b) := \{(x_i, y_i) \in S_m \mid V_i \leq w_i^*/b\} \tag{4}$$

from $Q_X$. The expected number of samples is $\mathbb{E}[N] = m/b$; thus, rejection sampling is more effective when the proposal distribution is similar to the target.

**Rejection sampling Clopper-Pearson (RSCP) bound.** Given $T_N$, our algorithm uses the CP bound to construct a PAC prediction set $C$ for $Q$. Let $\sigma_i := \mathbb{1}(V_i \leq w_i^*/b)$ indicate whether example $(x_i, y_i) \in S_m$ is accepted—i.e., $T_N(S_m, V, \vec{w}^*, b) = \{(x_i, y_i) \in S_m \mid \sigma_i = 1\}$, where $|T_N(S_m, V, \vec{w}^*, b)| = \sum_{i=1}^{m} \sigma_i$. Then, it follows that $U_{\text{RSCP}}$ bounds the error on $Q$, where

$$U_{\text{RSCP}}(C, S_m, V, \vec{w}^*, b, \delta) := U_{\text{CP}}(C, T_N(S_m, V, \vec{w}^*, b), \delta). \tag{5}$$

Thus, we have the following (see Appendix D.3 for the proof, and Algorithm 4 in Appendix E for a detailed description of the PS-R algorithm that leverages this bound):

**Theorem 2** *Define $U_{RSCP}$ as in (5). Let $\hat{\tau}$ be the solution of the following problem:*

$$\hat{\tau} = \max_{\tau \in \mathcal{T}} \tau \quad \text{subj. to} \quad U_{RSCP}(C_\tau, S_m, V, \vec{w}^*, b, \delta) \leq \varepsilon. \tag{6}$$

*Then, we have $\mathbb{P}_{S_m \sim P^m, V \sim U^m} \left[ L_Q(C_\tau) \leq \varepsilon \right] \geq 1 - \delta$ for any $\tau \leq \hat{\tau}$.*

Note that $V$ is sampled only once for Algorithm 4, so the randomness in $V$ can contribute to the failure of the correctness guarantee; however, the failure rate is controlled by $\delta$.

---

[4]In practice, we can improve stability of importance weights by considering source and target distributions induced by a feature mapping, e.g., learned using unsupervised domain-adaptation (Park et al., 2020b).

### 3.3 Approximate Importance Weights

So far, we have assumed that the true importance weight $w^*(x)$ is known. Since in practice, it needs to be estimated, we relax this assumption to only needing an uncertainty set of possible importance weights. This allows us to handle estimation error in the weights.

**Problem.** Let $S_m^X$ be unlabeled calibration examples from the source distribution (i.e., the covariates in $S_m$), $T_n^X$ be $n$ unlabeled calibration examples from the target distribution (denoted by $T_n^X \sim Q_X^n$), and $\vec{w}^* := (w_1^*, ..., w_m^*) \in \mathbb{R}^m$ be the vector of true importance weights $w_i^* := w^*(x_i)$, for $(x_i, y_i) \in S_m$. Then, we assume an uncertainty set $\mathcal{W} \subseteq \mathbb{R}^m$ that has $\vec{w}^*$ with high probability, i.e.,

$$\mathbb{P}_{S_m^X \sim P_X^m, T_n^X \sim Q_X^n} [\vec{w}^* \in \mathcal{W}] \geq 1 - \delta_w, \tag{7}$$

where $\delta_w \in (0, 1)$. We assume $\mathcal{W}$ has the form

$$\mathcal{W} := \{ w \in \mathbb{R}^m \mid \forall i \in [m] , \ \underline{w}_i \leq w_i \leq \overline{w}_i \},$$

for some $\underline{w}_i$ and $\overline{w}_i$. See the discussion on our choice of $\mathcal{W}$ in Appendix C.1.

**Robust Clopper-Pearson bound.** To construct a PAC prediction set $C$ for $Q$, it suffices to bound the worst-case error over $w \in \mathcal{W}$, i.e., we have the following (see Appendix D.4 for the proof):

**Theorem 3** *Suppose $\mathcal{W}$ satisfies (7). Define $U_{RSCP}$ as in (5). Let $\hat{\tau}$ be the solution of the following:*

$$\hat{\tau} = \max_{\tau \in \mathcal{T}} \tau \quad \text{subj. to} \quad \max_{w \in \mathcal{W}} U_{RSCP}(C_\tau, S_m, V, w, b, \delta_C) \leq \varepsilon. \tag{8}$$

*Then, we have $\mathbb{P}_{S_m \sim P^m, V \sim U^m, T_n^X \sim Q_X^n}[L_Q(C_\tau) \leq \varepsilon] \geq 1 - \delta_C - \delta_w$ for any $\tau \leq \hat{\tau}$.*

A key challenge applying Theorem 3 is solving the maximum over $w \in \mathcal{W}$. We propose a simple greedy algorithm that achieves the maximum.

**Greedy algorithm for robust $U_{RSCP}$.** The RSCP bound $U_{RSCP}$ satisfies certain monotonicity properties that enable us to efficiently compute an upper bound to the maximum in (8). In particular, if $C$ makes an error on $(x_i, y_i)$ (i.e., $y_i \notin C(x_i)$), then $U_{RSCP}$ is monotonically non-decreasing in $w_i^* = w^*(x_i)$; intuitively, this holds since a larger $w_i^*$ increases the probability that $(x_i, y_i)$ is included in $T_N(S_m, V, w^*, b)$, which in turn increases the empirical error $\bar{L}_{T_N(S_m, V, w^*, b)}(C)$. Conversely, if $C$ does not make an error on $(x_i, y_i)$ (i.e., $y_i \in C(x_i)$), $U_{RSCP}$ is non-increasing in $w_i^*$. More formally, we have the following result (see Appendix D.5 for a proof):

**Lemma 1** *For any $i \in [m]$, $U_{RSCP}(C, S_m, V, w^*, b, \delta)$ is monotonically non-decreasing in $w_i^*$ if $y_i \notin C(x_i)$, and monotonically non-increasing in $w_i^*$ if $y_i \in C(x_i)$.*

Our greedy algorithm leverages the monotonicity of $U_{RSCP}$. In particular, given $\mathcal{W}$ and $C$, the choice

$$\hat{w} := (\hat{w}_1, \ldots, \hat{w}_m), \qquad \text{where} \qquad \hat{w}_i = \begin{cases} \overline{w}_i & \text{if } y_i \notin C(x_i) \\ \underline{w}_i & \text{if } y_i \in C(x_i) \end{cases} \quad (\forall i \in [m]) \tag{9}$$

is the maximum value over $w \in \mathcal{W}$ of the constraint $U_{RSCP}(C_\tau, S_m, V, w, b, \delta_C)$ in (8), i.e.,

$$\max_{w \in \mathcal{W}} U_{RSCP}(C, S_m, V, w, b, \delta) = U_{RSCP}(C, S_m, V, \hat{w}, b, \delta). \tag{10}$$

Thus, we have the following, which follows by (10) and the same argument as the Theorem 3:

**Theorem 4** *Suppose $\mathcal{W}$ satisfies (7). Define $\hat{w}_{\tau, S_m^X, T_n^X}$ as in (9), making the dependency on $\tau$, $S_m^X$, and $T_n^X$ explicit, and $U_{RSCP}$ as in (5). Let $\hat{\tau}$ be the solution of the following problem:*

$$\hat{\tau} = \max_{\tau \in \mathcal{T}} \tau \quad \text{subj. to} \quad U_{RSCP}(C_\tau, S_m, V, \hat{w}_{\tau, S_m^X, T_n^X}, b, \delta_C) \leq \varepsilon. \tag{11}$$

*Then, we have $\mathbb{P}_{S_m \sim P^m, V \sim U^m, T_n^X \sim Q_X^n}[L_Q(C_\tau) \leq \varepsilon] \geq 1 - \delta_C - \delta_w$ for any $\tau \leq \hat{\tau}$.* [5]

**Importance weight estimation.** In general, to estimate the importance weights (IWs), some assumptions on their structure are required (Kanamori et al., 2009; Cortes et al., 2008; Nguyen et al.,

---

[5] We assume $b$ is given for simplicity, but our approach estimates it; see Appendix C.2 for details.

---

**Algorithm 1** PS-W: an algorithm using the robust RSCP bound in (20)

---

 1: **procedure** PS-W$(S_m, T_n^X, f, g, \mathcal{T}, \varepsilon, \delta_w, \delta_C, K, E)$
 2:     $\mathcal{W} \leftarrow$ ESTIMATEIWS$(S_m^X, T_n^X, g, \delta_w, K, E)$                    ($\triangleright$) Compute an uncertainty set $\mathcal{W}$
 3:     $b \leftarrow \max_{i \in [K]} \overline{w}_i$                    ($\triangleright$) Compute the maximum IW (see (19) in Appendix C.2)
 4:     **return** PS-ROBUST$(S_m, f, \mathcal{T}, \mathcal{W}, b, \varepsilon, \delta_C)$
 5: **procedure** ESTIMATEIWS$(S_m^X, T_n^X, g, \delta_w, K, E)$
 6:     Construct bins $B_1, \ldots, B_K$ using $g$ as described in Appendix B.2
 7:     Construct $\mathcal{W}$ using $B_1, \ldots, B_K, S_m^X, T_n^X, g, \delta_w$ and, $E$ as described in Appendix B.1
 8:     **return** $\mathcal{W}$
 9: **procedure** PS-ROBUST$(S_m, f, g, \mathcal{T}, \mathcal{W}, b, \varepsilon, \delta_C)$
10:     $V \sim$ Uniform$([0, 1])^m$
11:     $\hat{\tau} \leftarrow 0$
12:     **for** $\tau \in \mathcal{T}$ **do**                    ($\triangleright$) Grid search in ascending order
13:         Construct $\hat{w}$ using (9) given $\tau, S_m$, and $\mathcal{W}$
14:         **if** $U_{\text{RSCP}}(C_\tau, S_m, V, \hat{w}, b, \delta_C) \leq \varepsilon$ **then**
15:             $\hat{\tau} \leftarrow \max(\hat{\tau}, \tau)$
16:         **else break**
17:     **return** $\hat{\tau}$

---

2010; Lipton et al., 2018). We use a cluster-based approach (Cortes et al., 2008). In particular, we heuristically partition the feature space of the score function $f$ into $K$ bins by using a probabilistic classifier $g$ that separates source and target examples, and then estimate the source and target covariate distributions $p(x)$ and $q(x)$ based on the smoothness assumption over the distributions, where the degree of the smoothness is parameterized by $E$. Then, we can construct the uncertainty set $\mathcal{W}$ that satisfies the specified guarantee in (7). See Appendix B for details.

**Algorithm.** Our algorithm, called PS-W, is detailed in Algorithm 1; it solves (20) and also performs importance weight estimation according to (17) in Appendix B. In particular, Algorithm 1 constructs a prediction set that satisfies the PAC guarantee in Theorem 4. See Section 4.1 for our choice of parameters (e.g., $K$, $E$, and grid search parameters).

## 4 EXPERIMENTS

We show the efficacy of our approach on rate and support shifts on DomainNet and ImageNet.

### 4.1 EXPERIMENTAL SETUP

**Models.** For each source-target distribution pair, we split each the labeled source data and unlabeled target data into train and calibration sets. We use a separate labeled test set from the target for evaluation. For each shift from source to target, we use a deep neural network score function $f$ based on ResNet101 (He et al., 2016), trained using unsupervised domain adaptation based on the source and target training sets. See Appendix F for details, including data split.

**Prediction set construction.** To construct our prediction sets, we first estimate IWs by training a probabilistic classifier $g$ using the source and target training sets. Next, we use $g$ to construct heuristic IWs $w(x) = 1/g(s = 1 \mid x) - 11$, where $s = 1$ if $x$ is from source. Then, we estimate the lower and upper bound of the true IWs using Theorem 5 with $E = 0.001$ and $K = 10$ bins (chosen to contain equal numbers of heuristic IWs), where we compute the lower and upper Clopper-Pearson interval using the source and target calibration sets. Furthermore, given a confidence level $\delta$, we use $\delta_C = \delta_w = \delta/2$. For the grid search in line 12 of Algorithm 1, we increase $\tau$ by $10^{-7}$ until the bound $U_{\text{RSCP}}$ exceeds $1.5\varepsilon$. Finally, we evaluate the prediction set error on the labeled target test set.

**Baselines.** We compare the proposed method in Algorithm 1 (**PS-W**) with the following:

- **PS:** The prediction set using Algorithm 2 that satisfies the PAC guarantee from Theorem 1, based on Park et al. (2020a), which makes the i.i.d. assumption.
- **PS-C:** A Clopper-Pearson method addressing covariate shift by a conservative upper bound on the empirical loss (see Appendix E.2 for details), resulting in Algorithm 3 that solves the following:

$$\hat{\tau} = \arg\max_{\tau \in \mathcal{T}} \tau \quad \text{subj. to} \quad U_{\text{CP}}(C_\tau, S_m, \delta) \leq \varepsilon/b.$$

- **WSCI:** Weighted split conformal inference, proposed in (Tibshirani et al., 2019). Under the exchangeability assumption (which is somewhat weaker than our i.i.d. assumption), this approach

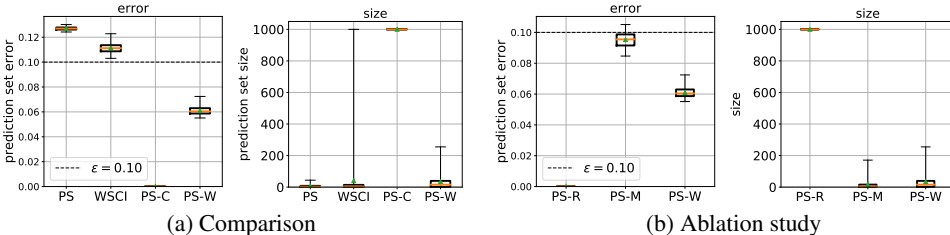

(a) Comparison                    (b) Ablation study

Figure 2: The prediction set error and size over 100 random trials under synthetic shift from ImageNet to ImageNet-C13. Parameters are $m = 20,000$, $\varepsilon = 0.1$, and $\delta = 10^{-5}$. (a) and (b) shows the box plots for comparison and ablation study, respectively.

provides correctness guarantees only for a single future test instance, i.e., it includes an $\varepsilon$ confidence level (denoted $\alpha$ in their paper) but not $\delta$. Also, it assumes the true IWs are known; following their experiments, we use the heuristic IWs constructed using $g$.

- **PS-R:** The prediction set using Algorithm 4 that satisfies the PAC guarantee from Theorem 2 with the heuristic IWs constructed using a probabilistic classifier $g$.
- **PS-M:** The prediction set using Algorithm 5, which is identical to PS-R except that PS-M estimates IWs heuristically using histogram density estimation and a calibration set. In particular, we partition the IW values into bins, project source and target calibration examples into bins based on the value of the probabilistic classifier $g$, and estimate the source density $\hat{p}_B$ and target density $\hat{q}_B$ for each bin. The estimated importance weight is $\hat{w}(x) = \hat{q}_B(x)/\hat{p}_B(x)$, where $\hat{p}_B$ and $\hat{q}_B$ are defined in (14) and (15), respectively.

**Metrics.** We measure performance via the prediction set error and size on a held-out test set—i.e., error is the fraction of $(x, y)$ such that $y \notin C(x)$ and size is the average of $|C(x)|$ over $x$.

**Rate shifts on DomainNet.** We consider settings where the model is trained on data from a variety of domains, but is then deployed on a specific domain; we call such a shift *rate shift*. For instance, a self-driving car may be trained on both day and night images, but tested during the night time. While the model should still perform well since the target is a subset of the source, the covariate shift can nevertheless invalidate prediction set guarantees. We consider rate shifts on DomainNet (Peng et al., 2019), which consists of images of $345$ classes from six domains (sketch, clipart, painting, quickdraw, real, and infograph); we use all domains as the source and each domain as a target.

**Support shifts on ImageNet.** Next, we consider support shifts, where the support of the target is different from the support of the source, but unsupervised domain adaptation is used to learn feature representations that align the two distributions (Ganin et al., 2016); then, the score function $f$ is trained on this representation. First, we consider ImageNet-C (Hendrycks & Dietterich, 2019), which modifies the original ImageNet dataset (Russakovsky et al., 2015) using 15 synthetic perturbations with 5 severity levels. We use 13 perturbations, omitting "snow" and "glass blur", which are computationally expensive to run. We consider the original ImageNet dataset as the source, and the all synthetic perturbations on all of ImageNet (denoted ImageNet-C13) as the target. See Appendix F.3 for data split. Second, we consider adversarial shifts, where we generate adversarial examples for ImageNet using the PGD attack (Madry et al., 2017) with $0.01$ $\ell_\infty$-norm perturbations with respect to a pretrained ResNet101. We consider the original ImageNet as the source and the adversarially perturbed ImageNet as the target.

## 4.2 Experimental Results

We summarize our results in Table 1, and provide more details in Figure 4 in Appendix G.2. Our PS-W algorithm satisfies the PAC constraint, and gives prediction sets with the smallest average normalized size among approaches that always satisfy the PAC constraint.

**Rate shifts on DomainNet.** We use $\varepsilon = 0.1$ and $\delta = 10^{-5}$. As can be seen in Table 1, the prediction set error of our approach (PS-W) does not violate the error constraint $\varepsilon = 0.1$, while all other comparing approaches (except for PS-C) violate it for at least one of the shifts. While PS-C satisfies the desired bound, it is overly conservative; its prediction sets are significantly larger than necessary, making it less useful for uncertainty quantification. For the shift to Infograph in Table 1, PS-W achieves relatively larger average prediction set size compared to other shifts; this is because the classification error of the score function $f$ over Infograph is large—71.28%, whereas

Table 1: Average prediction set error and sizes over 100 random trials under rate shifts on DomainNet (first six shifts) and support shifts on ImageNet (last two shifts). We denote an approach satisfying the $\varepsilon$ error constraint by ✓, and ✗ otherwise. The "normalized size" is the size divided by the total number of classes (i.e., 345 for DomainNet and 1000 for ImageNet). Parameters are $m = 50,000$ for DomainNet, $m = 20,000$ for ImageNet, $\varepsilon = 0.1$, and $\delta = 10^{-5}$. Our approach PS-W satisfies the $\varepsilon$ constraint, while producing prediction sets with the smallest average normalized size among approaches that always satisfy the error constraint. See Appendix G.2 for box plots.

| Shift | Baselines | | | | | | Ablations | | | | Ours | |
|---|---|---|---|---|---|---|---|---|---|---|---|---|
| | **PS** | | **WSCI** | | **PS-C** | | **PS-R** | | **PS-M** | | **PS-W** | |
| | error | size | error | size | error | size | error | size | error | size | error | size |
| All | ✓ (0.094) | 10.5 | ✓ (0.099) | 9.5 | ✓ (0.093) | 10.7 | ✓ (0.094) | 10.6 | ✓ (0.094) | 10.8 | ✓ (0.070) | 17.0 |
| Sketch | ✗ (0.142) | 13.1 | ✗ (0.116) | 18.6 | ✓ (0.020) | 141.7 | ✓ (0.097) | 28.2 | ✗ (0.105) | 26.1 | ✓ (0.078) | 40.3 |
| Painting | ✗ (0.159) | 15.4 | ✗ (0.113) | 30.0 | ✓ (0.025) | 125.4 | ✓ (0.096) | 37.7 | ✗ (0.103) | 34.5 | ✓ (0.076) | 52.8 |
| Quickdraw | ✓ (0.069) | 5.9 | ✓ (0.097) | 3.8 | ✓ (0.021) | 23.8 | ✓ (0.088) | 4.3 | ✓ (0.087) | 4.2 | ✓ (0.067) | 6.1 |
| Real | ✓ (0.079) | 8.7 | ✓ (0.087) | 7.2 | ✓ (0.032) | 47.8 | ✓ (0.080) | 8.7 | ✓ (0.087) | 7.1 | ✓ (0.068) | 11.8 |
| Clipart | ✗ (0.105) | 10.2 | ✗ (0.101) | 10.9 | ✓ (0.000) | 345.0 | ✓ (0.080) | 19.4 | ✓ (0.086) | 14.8 | ✓ (0.060) | 25.7 |
| Infograph | ✗ (0.363) | 36.4 | ✗ (0.114) | 165.1 | ✓ (0.000) | 345.0 | ✓ (0.085) | 202.6 | ✗ (0.107) | 177.4 | ✓ (0.078) | 216.4 |
| ImageNet-PGD | ✓ (0.090) | 5.5 | ✓ (0.096) | 4.7 | ✓ (0.000) | 1000.0 | ✓ (0.000) | 1000.0 | ✓ (0.074) | 7.8 | ✓ (0.049) | 13.9 |
| ImageNet-C13 | ✗ (0.127) | 9.3 | ✗ (0.111) | 67.0 | ✓ (0.000) | 1000.0 | ✓ (0.000) | 1000.0 | ✓ (0.095) | 15.9 | ✓ (0.061) | 35.8 |
| mean normalized size | – | | – | | 0.0338 | | 0.0257 | | – | | **0.0047** | |

that over Sketch is $37.16\%$. Even with the poor score function, our proposed approach still satisfies the $\varepsilon = 0.1$ error constraint. Finally, while our approach PS-W generally performs best subject to satisfying the error constraint, we note that our ablations PS-R and PS-M also perform well, providing different tradeoffs. First, the performance of PS-W is significantly more reliable than PS-R, but in some cases PS-R performs better (e.g., it produces slightly smaller prediction sets on rate shifts but significantly larger sets on support shifts). Alternatively, PS-M consistently produces smaller prediction sets, though it sometimes violates the $\varepsilon$ error constraints.

**Support shifts on ImageNet.** We show results for synthetic and adversarial shifts in Table 1 (and Figure 4 in Appendix G.2). As can be seen, the error of our approach (PS-W) is below the desired level. PS-R performs poorly, likely due to the uncalibrated point IW estimation—we find that calibrated importance weights mitigate these issues, though accounting for uncertainty in the IWs is necessary for achieving the desired error rate; see Appendix G.3 for details.

For adversarial shifts, the target classification error of the source-trained ResNet101 and the domain-adapted ResNet101 is $99.97\%$ and $28.05\%$, respectively. Domain adaptation can significantly decrease average error rate, but label predictions still do not have guarantees. However, our prediction set controls the prediction set error rate as specified by $\varepsilon$. As shown in Figure 4, our prediction set function outputs a prediction set for a given example that includes the true label at least $90\%$ of the time. Thus, downstream modules can rely on this guarantee for further decision-making.

**Ablation and sensitivity study.** We conduct an ablation study on the effect of IW calibration and the smoothness parameter $E$ for PS-W. We observe that the IW calibration considering uncertainty intervals around calibrated IWs is required for the PAC guarantee (e.g., Figure 2b). Also, we find that a broad range of $E$-s can be used to satisfy the PAC guarantee; we believe this result is due to the use of a domain adapted score function. See Appendices G.3 & G.5 for details.

## 5 CONCLUSION

We propose a novel algorithm for building PAC prediction sets under covariate shift; it leverages rejection sampling and the Clopper-Pearson interval, and accounts for uncertain IWs. We demonstrate the efficacy of our approach on natural, synthetic, and adversarial covariate shifts on DomainNet and ImageNet. Future work includes providing optimality guarantees on the prediction set size, rigorous estimation of the hyperparameter $E$, and incorporating probabilistic IW uncertainty estimates.

**Ethics statement.** We do not foresee any significant ethical issues with our work. One possible issue is that end-users may overly trust the prediction sets if they do not understand the limitations of our approach—e.g., it is only a high probability guarantee.

**Reproducibility statement.** Algorithms used for evaluation, including ours, are stated in Algorithm 1, Algorithm 2, Algorithm 3, Algorithm 4, and Algorithm 5, along with hyperparameters for algorithms in Section 4.1 and Appendix F. Related code is released[6]. The proof of our theorems are stated in Appendix D; the required assumption is stated in Assumption 1.

## ACKNOWLEDGMENTS

This work was supported in part by ARO W911NF-20-1-0080, AFRL and DARPA FA8750-18-C-0090, NSF award CCF 1910769, and NSF award 2031895 on the Mathematical and Scientific Foundations of Deep Learning (MoDL). Any opinions, findings and conclusions or recommendations expressed in this material are those of the authors and do not necessarily reflect the views of the Air Force Research Laboratory (AFRL), the Army Research Office (ARO), the Defense Advanced Research Projects Agency (DARPA), or the Department of Defense, or the United States Government. The authors are grateful to Art Owen for helpful comments.

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

# A   ADDITIONAL RELATED WORK

**Prediction sets under i.i.d. assumption.** We built our proposed approach on the known PAC prediction set approach (Wilks, 1941; Park et al., 2020a) due to its simplicity and sample efficiency. As other candidates, nested conformal prediction (Vovk et al., 2005; Gupta et al., 2021) reinterprets multiple known conformal prediction approaches using a scalar parameteriztion of prediction sets; but different from Wilks (1941) and Park et al. (2020a), it is not known to provide PAC guarantees. Kivaranovic et al. (2020) provides a PAC style guarantee, but their approach is limited to regression and is sample-inefficient, i.e., it requires a sample of size $O(1/\epsilon^2)$, while Park et al. (2020a) has a better sample complexity, as demonstrated in their paper.

**Prediction sets under various settings.** Prior prediction set algorithms have been considered in several settings. First, traditional conformal prediction (Vovk et al., 2005) often considers the setting where labeled examples arrive sequentially from the same distribution; there has also been work extending conformal prediction to the setting where the distribution is time-varying (Politis, 2015; Chernozhukov et al., 2018; Xu & Xie, 2021; Gibbs & Candès, 2021). Alternatively, there has been work on constructing risk-controlling prediction sets for supervised learning setting (Vovk, 2013; Park et al., 2020a; Bates et al., 2021; Angelopoulos et al., 2021). Finally, there has been recent work on conformal prediction in the meta-learning setting (Fisch et al., 2021); in particular, given a few labeled examples drawn from the new task, their approach leverages labeled examples from previous tasks to construct a conformal predictor for the new task, assuming the tasks are exchangeable.

**Developments after our work.** After our work was made publicly available, Jin et al. (2021) has developed a different, robust conformal inference approach to constructing prediction sets with estimated weights under covariate shift. Their algorithm assumes given upper and lower bounds on the importance weights, and uses the worst-case quantile over all weights that satisfy the constraint to set the critical values. Further, Yang et al. (2022) have developed a doubly robust approach to construct prediction sets satisfying approximate marginal coverage under covariate shift (which can be robust to estimating the weights and per-covariate prediction error), leveraging semiparametric efficiency theory. Qiu et al. (2022) have developed a parallel approach for the PAC case.

**Calibration.** An alternative way to quantify uncertainty is *calibrated prediction* (e.g., Brier, 1950; Cox, 1958; Miller, 1962; Murphy, 1972; Lichtenstein et al., 1977; DeGroot & Fienberg, 1983; Guo et al., 2017, etc), which aims to ensure that among instances with a predicted confidence $p$, the model is correct a fraction $p$ of the time. Techniques have been proposed to re-scale predicted confidences to improve calibration (Platt, 1999; Guo et al., 2017; Zadrozny & Elkan, 2001; 2002; Kuleshov & Liang, 2015; Kuleshov et al., 2018; Malik et al., 2019); including ones with theoretical guarantees (Kumar et al., 2019; Park et al., 2021) and ones that handle covariate shift (Park et al., 2020b; Wang et al., 2020). There are also methods to rigorously test calibration, dating back to Cox (1958); Miller (1962), see e.g., Lee et al. (2022) for a recent approach. These approaches provide a qualitatively different form of uncertainty quantification compared to the one we consider.

**Rejection sampling.** Rejection sampling (sometimes accept-reject sampling) is a well-known technique (Owen, 2013; Rubinstein & Kroese, 2016) dating back at least to von Neumann (1951). We highlight that most work on covariate shift relies on importance weighting; our rejection sampling approach is relatively less common and more novel. In fact, to the best of our knowledge, only Pagnoni et al. (2018) has used rejection sampling for a different problem in this area.

**IW estimation.** There has been a long line of work studying the problem of estimating importance weights (IWs), also called likelihood ratios, in a way that provides theoretical guarantees. For instance, (Nguyen et al., 2010) provides a convergence rate analysis of IW estimators—under smoothness assumptions, they provide a finite sample bound on the Hellinger distance between the true IW and estimated IW. Next, (Kanamori et al., 2009) shows a similar finite sample guarantee, assuming the true IW can be represented as a linear combination of kernels. Finally, (Cortes et al., 2008) proposes non-parametric IW estimators, modeling the source and target distribution by histograms over clusters in sample space.

The IW estimation approaches can be used in conjuction with prediction set construction. Compared to Candès et al. (2021), which only guarantees asymptotic validity under certain assumptions on the estimated IWs, Theorem 4 provides a finite-sample correctness guarantee. Furthermore, we explicitly describe an algorithm for approximate IWs, required by Theorem 4, in Appendix B.

## B   IMPORTANCE WEIGHT ESTIMATION

In general, to estimate the importance weights (IWs), some assumptions on their structure are required. A number of approaches have been proposed, with varying guarantees under different assumptions (Kanamori et al., 2009; Cortes et al., 2008; Nguyen et al., 2010; Lipton et al., 2018). We use a cluster-based approach (Cortes et al., 2008); our approach is compatible with any of these strategies if they can be modified to provide uncertainty estimates of the IWs.

In our approach, given a partition $\mathcal{X} = \bigcup_{j=1}^{K} B_j$ into bins, we can estimate the IWs based on the fractions of source and target samples in each bin. If the partition is sufficiently fine, then we can obtain confidence intervals around the estimated IWs with finite-sample guarantees. However, this strategy requires the number of bins in the partition to be exponential in the dimension of $\mathcal{X}$. Thus, in practice, we use a heuristic to construct the partition. We describe the cluster-based approach and our partition construction heuristic below.

### B.1   CLUSTER-BASED APPROACH

We assume given unlabeled calibration sets $S_m^X$ and $T_n^X$, where $S_m^X$ consists of i.i.d. samples $x_i \sim p(x)$ for $i \in [m]$, and $T_n^X$ consists of i.i.d. samples $x_i \sim q(x)$ for $i \in [n]$, respectively[7]. Roughly speaking, the cluster-based strategy estimates the average IW in each bin $B_j$; assuming $p$ and $q$ are roughly constant in each bin, these accurately estimate the true IWs. Let $j(x)$ be the bin containing $x$ (i.e., $x \in B_{j(x)}$), and let

$$p_B(x) := p_{j(x)} \text{ s.t. } p_j = \int_{B_j} p(x') \, \mathrm{d}x' \quad \text{and} \quad q_B(x) := q_{j(x)} \text{ s.t. } q_j = \int_{B_j} q(x') \, \mathrm{d}x'$$

be the (unnormalized) approximations of the densities $p$ and $q$, respectively, that are constant on each bin. We assume that $p_B$ and $q_B$ are accurate approximations:

**Assumption 1** Given $E \in \mathbb{R}_{\geq 0}$, the partition satisfies

$$\int_{B_j} |p(x) - p(x')| \mathrm{d}x' \leq E \quad \text{and} \quad \int_{B_j} |q(x) - q(x')| \mathrm{d}x' \leq E \qquad (j \in [K], \forall x \in B_j). \quad (12)$$

Thus, $p$ and $q$ are roughly constant on the partitions. In general, (12) can hold for any $E \in \mathbb{R}_{>0}$ if $p$ and $q$ are Lipschitz continuous and each $B_j$ is sufficiently small (see Appendix B.3 for discussion). Then, under Assumption 1, it can be verified that

$$|v(x) \cdot p(x) - p_B(x)| \leq E \qquad \text{and} \qquad |v(x) \cdot q(x) - q_B(x)| \leq E \qquad (\forall x \in \mathcal{X}), \qquad (13)$$

where $v(x) = v_{j(x)}$, and $v_j = \int_{B_j} \mathrm{d}x'$ is the volume of bin $B_j$ (see the proof of Theorem 5 for the validity of (13)). Next, we have the following empirical estimates of $p_B$ and $q_B$, respectively:

$$\hat{p}_B(x) := \hat{p}_{j(x)} \text{ s.t. } \hat{p}_j = \frac{1}{m} \sum_{x' \in S_m^X} \mathbb{1}\left(x' \in B_j\right) \text{ and} \qquad (14)$$

$$\hat{q}_B(x) := \hat{q}_{j(x)} \text{ s.t. } \hat{q}_j = \frac{1}{n} \sum_{x' \in T_n^X} \mathbb{1}\left(x' \in B_j\right). \qquad (15)$$

Now, $\mathbb{1}(x' \in B_j)$ has distribution Bernoulli$(p_j)$ when $x \sim P$, thus $m \cdot \hat{p}_j$ has distribution Binom$(m, p_j)$, and $p_j$ is contained in a Clopper-Pearson interval around $\hat{p}_j$ with high probability; in particular, let $\underline{\theta}$ be the Clopper-Pearson lower bound corresponding to the Clopper-Pearson upper bound defined in Section 2.2, i.e., $\mathbb{P}_{k \sim \text{Binom}(m,\mu)}[\mu \geq \underline{\theta}(k; m, \delta)] \geq 1 - \delta$. Then, we have

$$\underline{\theta}(m \cdot \hat{p}_j; m, \delta') \leq p_j \leq \overline{\theta}(m \cdot \hat{p}_j; m, \delta') \qquad (16)$$

with probability at least $1 - \delta'$ with respect to the samples $S_m^X$. Combining (13) and (16), we have the following result (and see Appendix D.6 for a proof):

---

[7] We can use the same calibration set to construct IWs and prediction sets due to the union bound in Theorem 3.

**Theorem 5** *Letting* $\delta' = \delta_w/(2K)$ *and* $[v]^+ := \max\{0, v\}$ *for all* $v \in \mathbb{R}$, *we have*

$$\underline{w}(x) := \frac{[\underline{\theta}\left(n \cdot \hat{q}_B(x); n, \delta'\right) - E]^+}{\overline{\theta}\left(m \cdot \hat{p}_B(x); m, \delta'\right) + E} \leq w^*(x) \leq \overline{w}(x) := \frac{\overline{\theta}\left(n \cdot \hat{q}_B(x), n, \delta'\right) + E}{[\underline{\theta}\left(m \cdot \hat{p}_B(x), m, \delta'\right) - E]^+} \quad (\forall x \in \mathcal{X})$$

(17)

*with probability at least* $1 - \delta_w$ *over* $S_m^X$ *and* $T_n^X$.

We use these upper and lower bounds on the IWs as the inputs $\underline{w}$ and $\overline{w}$ to our algorithm—i.e.,

$$\mathcal{W} = \{w : \mathcal{X} \to \mathbb{R} \mid \forall x \in \mathcal{X}, \ \underline{w}(x) \leq w(x) \leq \overline{w}(x)\},$$

and use $b = \max_{x \in \mathcal{X}} \overline{w}(x) = \max_{j \in [K]} \overline{w}_j$ as the maximum IW. If $b$ is known, we need importance weights associated with source calibration samples $S_m^X$; thus we use a simpler form of $\mathcal{W}$ as follows:

$$\mathcal{W} = \{w \in \mathbb{R}^m \mid \forall i \in [m], \ \underline{w}_i \leq w_i \leq \overline{w}_i\},$$

where $\underline{w}_i := \underline{w}(x_i)$ and $\overline{w}_i := \overline{w}(x_i)$ for $x_i \in S_m^X$.

Theorem 5 under Assumption 1 is one way to construct uncertainty set $\mathcal{W}$ that contains the true IWs.

**Corollary 1** *Given* $K \in \mathbb{N}$, $E \in \mathbb{R}_{\geq 0}$, *and* $\delta_w \in (0, 1)$, *suppose Assumption 1 is satisfied and* $\mathcal{W}$ *is constructed using Theorem 5. Then, we have*

$$\mathbb{P}_{S_m^X \sim P_X^m, T_n^X \sim Q_X^n}\left[w^* \in \mathcal{W}\right] \geq 1 - \delta_w.$$

### B.2 PARTITION CONSTRUCTION HEURISTIC

In general, exponentially many bins are needed to guarantee Assumption 1. Instead, we consider an intuitive heuristic for constructing these bins, so that the importance weights $w(x)$—rather than the density functions $p(x)$ and $q(x)$ individually—are roughly constant on each bin, inspired by (Park et al., 2021; 2020b); a standard heuristic for estimating IWs is to train a probabilistic classifier $g(s \mid x)$ to distinguish source and target training examples, and then use these probabilities to construct the IWs. In particular, define the distribution

$$g^*(x, y) = \frac{1}{2}p(x) \cdot \mathbb{1}(s = 1) + \frac{1}{2}q(x) \cdot \mathbb{1}(s = 0).$$

Then, letting $g^*(y \mid x)$ be the conditional distribution, we have $w^*(x) = 1/g^*(s = 1 \mid x) - 1$ (Bickel et al., 2007). Thus, we train $g(s \mid x) \approx g^*(s \mid x)$ and construct bins according to $w(x) = 1/g(s = 1 \mid x) - 1$, i.e.,

$$B_j = \{x \in \mathcal{X} \mid w(x) \in [w_j, w_{j+1})\},$$

where $0 = w_1 \leq w_2 \leq ... \leq w_{K+1} = \infty$. Finally, we describe how to train $g$. Let $S_{m'}^X$ and $T_{n'}^X$ be unlabeled training examples from a source and target, respectively; then, the set

$$R_{m',n'}^X = \{(x, 1) \mid x \in S_{m'}^X\} \cup \{(x, 0) \mid x \in T_{n'}^X\}$$

consists of i.i.d. samples $(x, y) \sim s^*(x, y)$. Thus, we can train $s$ on $R_{m',n'}^X$ using supervised learning. In practice, the corresponding IW estimates $w(x)$ can be inaccurate partly since $w$ is likely overfit to $R_{m',n'}^X$, which is why re-estimating the IWs in each bin according to Theorem 5 remains necessary.

### B.3 DENSITY ESTIMATION SATISFYING ASSUMPTION 1

We consider the following binning strategy that satisfies Assumption 1 for Lipschitz continuous PDFs. In particular, assume the PDFs $p(x)$ and $q(x)$ are $L$-Lipschitz continuous for some norm $\|\cdot\|$. Then, for any given $E$, construct each bin $B_j$ such that

$$\|x - x'\| \leq \frac{E}{L \cdot v_j} \qquad \forall x, x' \in B_j,$$

where $v_j$ is the volume of bin $B_j$. Then, we know that

$$|p(x) - p(x')| \leq L\|x - x'\| \leq \frac{E}{v_j},$$

so

$$\int_{B_j} |p(x) - p(x')| \mathrm{d}x' \leq E.$$

Similarly, we have

$$\int_{B_j} |q(x) - q(x')| \mathrm{d}x' \leq E.$$

Thus, the bins satisfy Assumption 1. Finally, assuming $\| \cdot \|$ is the $L_\infty$ norm, we describe one way to construct bins. In particular, construct bins by taking an $\varepsilon$-net with $\varepsilon = (E/L)^{1/(d+1)}$, where $d$ is the dimension of $\mathcal{X}$. Then, we have

$$\|x - x'\| \leq \varepsilon = \frac{\varepsilon^{d+1}}{\varepsilon^d} = \frac{E}{L \cdot v_j},$$

as desired.

## C ADDITIONAL DISCUSSION ON IMPORTANCE WEIGHTS

### C.1 APPROXIMATE IMPORTANCE WEIGHTS

In Section 3.3, we assume $\mathcal{W}$ has the following form:

$$\mathcal{W} := \left\{ w \in \mathbb{R}^m \mid \underline{w}_i \leq w_i \leq \overline{w}_i \right\}.$$

However, considering the fact that the expected importance weight is one, i.e., $\mathbb{E}_{x \sim P}[w^*(x)] = 1$, the uncertainty set $\mathcal{W}$ that contains the true importance weights with high probability can be further constrained as follows:

$$\mathcal{W}' := \left\{ w \in \mathbb{R}^m \mid \forall i \in [m] , \ \underline{w}_i \leq w_i \leq \overline{w}_i , \ \underline{c} \leq \sum_{i=1}^m w_i \leq \overline{c} \right\}$$

for some $\underline{c}$ and $\overline{c}$. In particular, using the Hoeffding's inequality for example, we can estimate $\underline{c}$ and $\overline{c}$ such that $w_1, \ldots, w_m$ can be the part of the true importance weight $w^*$ that satisfies the mass constraint $\mathbb{E}_{x \sim P}[w^*(x)] = 1$.

Recall that we have to find a maximizer in the uncertainty set $\mathcal{W}'$ as in (8); however, due to the additional constraint on $\sum_{i=1}^m w_i$ in $\mathcal{W}'$, it is challenging to solve the maximization problem exactly.

### C.2 MAXIMUM IMPORTANCE WEIGHT ESTIMATION

To generalize our approach to estimate the maximum importance weight $b$, we redefine the uncertainty set $\mathcal{W}$ over importance weights as follows:

$$\mathcal{W} := \left\{ w : \mathcal{X} \to \mathbb{R} \mid \forall x \in \mathcal{X}, \ \underline{w}(x) \leq w(x) \leq \overline{w}(x) \right\}$$

for some $\underline{w} : \mathcal{X} \to \mathbb{R}$ and $\overline{w} : \mathcal{X} \to \mathbb{R}$ such that it contains the true importance weight $w^*$ with high probability—i.e.,

$$\mathbb{P}_{S_m^X \sim P_X^m, T_n^X \sim Q_X^n} [w^* \in \mathcal{W}] \geq 1 - \delta_w. \tag{18}$$

Given this, the maximum importance weight is obtained as follows:

$$\hat{b} = \max_{x \in \mathcal{X}} \overline{w}(x).$$

Considering that $\mathcal{W}$ can be estimated using binning as in Appendix B, the maximum importance weight is rewritten as follows:

$$\hat{b} = \max_{x \in \mathcal{X}} \overline{w}(x) = \max_{j \in [K]} \overline{w}_j, \tag{19}$$

where

$$\overline{w}_j := \frac{\overline{\theta}\left(n \cdot \hat{q}_j, n, \delta'\right) + E}{\left[\underline{\theta}\left(m \cdot \hat{p}_j, m, \delta'\right) - E\right]^+}.$$

Here, $\underline{\theta}, \overline{\theta}, m, n, \delta', E, \hat{p}_j$, and $\hat{q}_j$ are defined in Theorem 5 and Appendix B.

The guarantee (18) implies the guarantee (7). Thus, for the maximization over the uncertainty set in (8), we use the original definition $\mathcal{W}$ and the greedy algorithm for $\hat{w}$ in (9). Then, Theorem 4 with the estimated $b$ using (19) still holds as follows:

**Theorem 6** *Suppose Assumption 1 is satisfied and $\mathcal{W}$ is estimated using Theorem 5. Define $\hat{w}_{\tau, S_m^X, T_n^X}$ as in (9) and $\hat{b}_{S_m^X, T_n^X}$ as in (19), making the dependency on $\tau$, $S_m^X$, and $T_n^X$ explicit, and $U_{RSCP}$ as in (5). Let $\hat{\tau}$ be the solution of the following problem:*

$$\hat{\tau} = \max_{\tau \in \mathcal{T}} \tau \quad subj. \ to \quad U_{RSCP}(C_\tau, S_m, V, \hat{w}_{\tau, S_m^X, T_n^X}, \hat{b}_{S_m^X, T_n^X}, \delta_C) \leq \varepsilon. \tag{20}$$

*Then, we have $\mathbb{P}_{S_m \sim P^m, V \sim U^m, T_n^X \sim Q_X^n}[L_Q(C_\tau) \leq \varepsilon] \geq 1 - \delta_C - \delta_w$ for any $\tau \leq \hat{\tau}$.*

## C.3 CHOOSING HYPERPARAMETERS

In general, there is no systematic way to choose the smoothness parameter $E$ and the number of bins $K$; we briefly discuss strategies for doing so.

**Number of bins $K$.** The bins are defined in one dimensional space as described in Appendix B.2, so we follow the standard practice in the calibration literature for binning (Guo et al., 2017; Park et al., 2021), where $K$ is between 10 and 20. As we are using equal-mass binning, we choose the number of bins so that each bin contains sufficiently many source examples (in our case, 5000 examples) for the length of the Clopper-Pearson interval over IWs of each bin to be below some threshold (in our case, $10^{-3}$), which leads us to $K = 10$. We provide a sensitivity analysis in Appendix G.6.

**Smoothness parameter $E$.** Estimating $E$ (i.e., computing the integral of the difference of source and target probabilities) is intractable in general as it requires that we perform density estimation in a high-dimensional space (i.e., 2048 in our case), and then integrate this density over each bin. To avoid this hyperparameter selection, we choose $E$ equal to zero or close to zero (in our case, 0.001). Intuitively, we find that binning based on the source-discriminator scores is an effective way to group examples with similar IWs; thus, the main contribution to the uncertainty is the error of the point estimate. We provide a sensitivity analysis on $E$ in Appendix G.5. Importantly, note that PS-W even with $E = 0$ satisfies PAC criterion. One direction for future work is devising better strategies for choosing $E$.

## C.4 COMPARISON WITH WSCI

**Guarantee.** The main difference between WSCI and PS-W lies in the guarantee provided (rather than prediction set size); PS-W provides a stronger guarantee than WSCI. In particular, WSCI does not provide a PAC guarantee. Instead, to satisfy the coverage probability guarantee, it requires a new calibration set for every new test example. However, in practice, we usually have a single held-out calibration set. Thus, our approach PS-W provides a guarantee that holds conditioned on this set. This difference is illustrated in Figure 3a, which compares WSCI and PS-W given the true importance weight. PS-W produces a larger set size than WSCI, but strictly satisfies the error constraint. The shortcoming of WSCI can also be observed in Figure 2 in Tibshirani et al. (2019), which empirically shows that the guarantee only holds on average over examples.

**Usage of IWs.** WSCI requires the true IWs, whereas our method can use approximate IWs. Also, IWs are used differently. Given the true importance weights, WSCI uses the importance weights to reweight examples in the calibration set, and PS-W uses the importance weights to generate a target labeled calibration set using rejection sampling.

# D  PROOFS

## D.1  PROOF OF THEOREM 1

First, note that the constraint in (1) implies $F(\bar{L}_{S_m}(C_\tau); m, \varepsilon) \leq \delta$; conversely, any value of $\tau$ satisfying $F(\bar{L}_{S_m}(C_\tau); m, \varepsilon) \leq \delta$ also satisfies $\bar{L}_{S_m}(C_\tau) \leq k(m, \varepsilon, \delta)$. Thus, we can rewrite (1) as

$$\hat{\tau} = \underset{\tau \in \mathcal{T}}{\arg\max} \; \tau \quad \text{subj. to} \quad F(\bar{L}_{S_m}(C_\tau); m, \varepsilon) \leq \delta.$$

On the other hand, if $\tau$ satisfies (3), then by definition of $U_{\mathrm{CP}}(C_\tau, S_m, \delta) = \bar{\theta}(k; m, \delta)$, we have

$$\inf\{\theta \in [0, 1] \mid F(\bar{L}_{S_m}(C_\tau); m, \theta) \leq \delta\} \cup \{1\} \leq \varepsilon,$$

which implies that $F(\bar{L}_{S_m}(C_\tau); m, \varepsilon) \leq \delta$ (since the infimum is obtained within the set since the binomial CDF $F$ is continuous in $\varepsilon$). Conversely, any value of $\tau$ satisfying $F(\bar{L}_{S_m}(C_\tau); m, \varepsilon) \leq \delta$ also satisfies $U_{\mathrm{CP}}(C_\tau, S_m, \delta) \leq \varepsilon$. Thus, we can rewrite (3) as

$$\hat{\tau} = \underset{\tau \in \mathcal{T}}{\arg\max} \; \tau \quad \text{subj. to} \quad F(\bar{L}_{S_m}(C_\tau); m, \varepsilon) \leq \delta,$$

implying (1) and (3) are equal. Thus, the claim follows from Theorem 1 of (Park et al., 2020a) and the fact that the error $L_P(C_\tau)$ is monotonically increasing in $\tau$. $\square$

## D.2  MONOTONICITY OF THE CLOPPER-PEARSON BOUND

The CP bound $U_{\mathrm{CP}}$ enjoys certain monotonicity properties that we will need. Intuitively, the CDF decreases as the number of observations $m$ increases while holding the number of successes $k$ fixed, but increases if both $m$ and $k$ are increased by the same amount (i.e., holding the number of failures $m - k$ fixed). In particular, we have the following:

**Lemma 2** *We have* $\bar{\theta}(k; m - 1, \delta) \geq \bar{\theta}(k; m, \delta)$ *and* $\bar{\theta}(k - 1; m - 1, \delta) \leq \bar{\theta}(k; m, \delta)$.

*Proof.* Recall that $F(k; m, \theta)$ is the cumulative distribution function of a binomial distribution $\mathrm{Binom}(m, \theta)$, or equivalently of the random variable $\sum_{i=1}^{m} X_i$, where $X_i \sim \mathrm{Bernoulli}(\theta)$ are i.i.d.

**Decreasing case.** If $k \leq m - 1$, then we have

$$\sum_{i=1}^{m} X_i \leq k \Rightarrow \sum_{i=1}^{m-1} X_i \leq k,$$

hence

$$\mathbb{P}\left[\sum_{i=1}^{m} X_i \leq k\right] \subseteq \mathbb{P}\left[\sum_{i=1}^{m-1} X_i \leq k\right],$$

so $F(k; m, \theta) \leq F(k; m - 1, \theta)$.

Then, we have

$$\begin{aligned} \bar{\theta}(k; m, \delta) &:= \inf\{\theta \in [0, 1] \mid F(k; m, \theta) \leq \delta\} \cup \{1\} \\ &\leq \inf\{\theta \in [0, 1] \mid F(k; m - 1, \theta) \leq \delta\} \cup \{1\} \\ &=: \bar{\theta}(k; m - 1, \delta), \end{aligned}$$

thus $\bar{\theta}$ is monotonically non-increasing in $m$.

**Increasing case.** We have

$$\sum_{i=1}^{m-1} X_i \leq k - 1 \Rightarrow \sum_{i=1}^{m} X_i \leq k,$$

hence

$$\mathbb{P}\left[\sum_{i=1}^{m-1} X_i \leq k - 1\right] \subseteq \mathbb{P}\left[\sum_{i=1}^{m} X_i \leq k\right],$$

so $F(k-1; m-1, \theta) \leq F(k; m, \theta)$.

Then, we have

$$
\begin{aligned}
\overline{\theta}(k; m, \delta) &:= \inf \{\theta \in [0, 1] \mid F(k; m, \theta) \leq \delta\} \cup \{1\} \\
&\geq \inf \{\theta \in [0, 1] \mid F(k-1; m-1, \theta) \leq \delta\} \cup \{1\} \\
&=: \overline{\theta}(k-1; m-1, \delta),
\end{aligned}
$$

thus $\overline{\theta}$ is monotonically jointly non-decreasing in $(m, k)$. $\square$

### D.3 PROOF OF THEOREM 2

The rejection sampling prediction set consists of two steps: (i) generate target samples, using source samples $S_m$, importance weights $w$, and an upper bound on their maximum value $b$, and (ii) construct the Clopper-Pearson prediction set using the generated target samples.

From rejection sampling, we choose $N := \sum_{i=1}^{m} \sigma_i$ samples from $S_m$, denoting them by $T_N$; here, $N \sim \text{Binom}(m, 1/b)$, and $1/b$ is the acceptance probability (von Neumann, 1951)—i.e.,

$$
\mathbb{P}\left[V' \leq \frac{w(X)}{b}\right] = \frac{1}{b},
$$

where $V' \sim \text{Uniform}([0, 1])$. The samples in $T_N$ are independent and identically distributed, conditionally on the random number $N$ of samples being equal to any fixed value $n$. The reason is that one can view the rejection sampling algorithm proceeding in stages, iterating through the samples one by one. The first stage starts at the very beginning, and then each stage ends when a datapoint is accepted, followed by starting a new stage at the next datapoint. The last stage ends at the last datapoint.

Based only on the source samples observed in one stage, rejection sampling produces a sample from the target distribution. Thus, within each stage, we produce one sample from the target distribution, and because each stage is independent of all the other ones, conditionally on any number of stages reached, our produced target samples are iid. Thus, we can use the Clopper-Pearson bound conditionally on each $N = n$.

To this end, let $\hat{\tau}(S_m, V) = \hat{\tau}$ to explicitly denote the dependence on $S_m$ and $V$, and let

$$
\tilde{\tau}(T_n) = \arg\max_{\tau \in \mathcal{T}} \tau \quad \text{subj. to} \quad U_{\text{RSCP}}(C_\tau, T_n, \delta) \leq \varepsilon.
$$

Note that conditioned on obtaining $n$ samples using rejection sampling (i.e., $|T_n(S_m, V, w, b)| = n$), we have $\hat{\tau}(S_m, V) \overset{D}{=} \tilde{\tau}(T_n)$, where $\overset{D}{=}$ denotes equality in distribution. Then, we have

$$
\begin{aligned}
\mathbb{P}_{S_m \sim P^m, V \sim U^m} &\left[L_Q(C_{\hat{\tau}(S_m, V)}) \leq \varepsilon\right] \\
&= \sum_{n=0}^{m} \mathbb{P}_{S_m \sim P^m, V \sim U^m}[L_Q(C_{\hat{\tau}(S_m, V)}) \leq \varepsilon \mid N = n] \cdot \mathbb{P}[N = n] \\
&= \sum_{n=0}^{m} \mathbb{P}_{T_n \sim Q^n}[L_Q(C_{\tilde{\tau}(T_n)}) \leq \varepsilon] \cdot \mathbb{P}[N = n] \\
&\geq \sum_{n=0}^{m} (1 - \delta) \cdot \mathbb{P}[N = n] \\
&= 1 - \delta,
\end{aligned}
$$

where the inequality follows by Theorem 1. The claim follows. $\square$

### D.4 PROOF OF THEOREM 3

First, let

$$
\tilde{\tau} = \arg\max_{\tau \in \mathcal{T}} \tau \quad \text{subj. to} \quad U_{\text{RSCP}}(C_\tau, S_m, V, \vec{w}^*, b, \delta_C) \leq \varepsilon, \tag{21}
$$

which satisfies $\mathbb{P}_{S_m \sim P^m, V \sim U^m} [L_Q(C_{\tilde{\tau}}) \leq \varepsilon] \geq 1 - \delta_C$ by Theorem 2. Now, with probability at least $1 - \delta_w$, we have $\vec{w}^* \in \mathcal{W}$. Under this event, we have

$$U_{\mathrm{RSCP}}(C_\tau, S_m, V, \vec{w}^*, b, \delta_C) \leq \max_{w \in \mathcal{W}} U_{\mathrm{RSCP}}(C_\tau, S_m, V, w, b, \delta_C),$$

so $\hat{\tau}$ satisfies the constraint in (21). Thus, we must have $\hat{\tau} \leq \tilde{\tau}$. By monotonicity of $L_Q(C_\tau)$ in $\tau$, we have $L_Q(C_{\hat{\tau}}) \leq L_Q(C_{\tilde{\tau}})$, which implies that

$$\mathbb{P}_{S_m \sim P^m, V \sim U^m, T_n^X \sim Q_X^n} [L_Q(C_{\hat{\tau}}) \leq \varepsilon] \geq \mathbb{P}_{S_m \sim P^m, V \sim U^m} [L_Q(C_{\tilde{\tau}}) \leq \varepsilon] \geq 1 - \delta_C,$$

where the last step follows by Theorem 2. The claim follows by a union bound, since $\vec{w}^* \in \mathcal{W}$ with probability at least $1 - \delta_w$. $\square$

### D.5 PROOF OF LEMMA 1

Let $w$ and $v$ be IWs where $w(x_i) \geq v(x_i)$ and $w(x_j) = v(x_j)$ for $j \neq i$. Additionally, we use the following shorthands:

$$n_w := \sum_{i=1}^m \mathbb{1}\left( V_i \leq \frac{w(x_i)}{b} \right),$$

$$T_{n_w} := \left\{ (x_i, y_i) \in S_m \,\middle|\, V_i \leq \frac{w(x_i)}{b} \right\},$$

$$k_w := \sum_{(x,y) \in T_{n_w}} \mathbb{1}\left( y \notin C(x) \right),$$

$$n_v := \sum_{i=1}^m \mathbb{1}\left( V_i \leq \frac{v(x_i)}{b} \right),$$

$$T_{n_v} := \left\{ (x_i, y_i) \in S_m \,\middle|\, V_i \leq \frac{v(x_i)}{b} \right\}, \text{ and}$$

$$k_v := \sum_{(x,y) \in T_{n_v}} \mathbb{1}\left( y \notin C(x) \right).$$

Here, $n_w \geq n_v$ since $w(x_i) \geq v(x_i)$. Finally, recall that $F(k; m, \theta)$ be the cumulative distribution function of a binomial random variable $\sum_{i=1}^m X_i$, where $X_i \sim \mathrm{Bern}(\theta)$.

**Non-decreasing case.** If $y_i \notin C(x_i)$, there are two cases to consider:

1. If $\frac{v(x_i)}{b} < V_i \leq \frac{w(x_i)}{b}$, then we can verify that $n_w = n_v + 1$ and $k_w = k_v + 1$.
2. Otherwise, we can verify that $n_w = n_v$ and $k_w = k_v$.

In both cases, $k_w \geq n_v$ and $n_w \geq n_v$. Since $\bar{\theta}$ is monotonically jointly non-decreasing in $(m, k)$ as in Lemma 2, we have

$$\begin{aligned}
U_{\mathrm{RSCP}}(C, S_m, V, w, b, \delta) &:= U_{\mathrm{CP}}(\bar{L}_{T_{n_w}}(C), \delta) \\
&:= \bar{\theta}(k_w; n_w, \delta) \\
&\geq \bar{\theta}(k_v; n_v, \delta) \\
&=: U_{\mathrm{CP}}(\bar{L}_{T_{n_v}}(C), \delta) \\
&=: U_{\mathrm{RSCP}}(C, S_m, V, v, b, \delta),
\end{aligned}$$

thus $U_{\mathrm{RSCP}}$ is monotonically non-decreasing in $w(x_i)$.

**Non-increasing case.** If $y_i \in C(x_i)$, then $k_w = k_v$. Since $\bar{\theta}$ is monotonically non-increasing in $m$ as in Lemma 2, we have

$$\begin{aligned}
U_{\mathrm{RSCP}}(C, S_m, V, w, b, \delta) &:= U_{\mathrm{CP}}(\bar{L}_{T_{n_w}}(C), \delta) \\
&:= \bar{\theta}(k_w; n_w, \delta) \\
&\leq \bar{\theta}(k_v; n_v, \delta) \\
&=: U_{\mathrm{CP}}(\bar{L}_{T_{n_v}}(C), \delta) \\
&=: U_{\mathrm{RSCP}}(C, S_m, V, v, b, \delta),
\end{aligned}$$

thus $U_{\text{RSCP}}$ is monotonically non-increasing in $w(x_i)$.

## D.6    PROOF OF THEOREM 5

Recall that

$$\hat{p}_B(x) := \sum_{j=1}^{K} \mathbb{1}\left(x \in B_j\right) \left[\frac{1}{m} \sum_{x' \in S_m^X} \mathbb{1}\left(x' \in B_j\right)\right],$$

$$\hat{q}_B(x) := \sum_{j=1}^{K} \mathbb{1}\left(x \in B_j\right) \left[\frac{1}{n} \sum_{x' \in T_n^X} \mathbb{1}\left(x_i \in B_j\right)\right],$$

$$p_B(x) := \sum_{j=1}^{K} \mathbb{1}\left(x \in B_j\right) \int_{B_j} p(x')\, \mathrm{d}x',$$

$$q_B(x) := \sum_{j=1}^{K} \mathbb{1}\left(x \in B_j\right) \int_{B_j} q(x')\, \mathrm{d}x', \text{ and}$$

$$v(x) := v_{j(x)} = \int_{B_{j(x)}} \mathrm{d}x'.$$

Due to the assumption of (12), $|v(x) \cdot p(x) - p_K(x)|$ is bounded for any $x \in B_j$ as follows:

$$\begin{aligned} |v(x) \cdot p(x) - p_B(x)| = \left|\int_{B_j} p(x)\, \mathrm{d}x' - \int_{B_j} p(x')\, \mathrm{d}x'\right| &= \left|\int_{B_j} p(x) - p(x')\mathrm{d}x'\right| \\ &\leq \int_{B_j} |p(x) - p(x')|\, \mathrm{d}x' \\ &= E. \end{aligned} \tag{22}$$

Similarly,

$$|v(x) \cdot q(x) - q_B(x)| \leq E. \tag{23}$$

Observe that $m\hat{p}(x) \sim \text{Binom}\left(m, \int_{B_j} p(x')\mathrm{d}x'\right)$ for any $x \in B_j$; thus $p_K$ is bounded with probability at least $1 - \delta'$ as follows due to the Clopper-Pearson interval $(\underline{\theta}, \overline{\theta})$:

$$\underline{\theta}(m\hat{p}(x); m, \delta') \leq p_K(x) \leq \overline{\theta}(m\hat{p}(x); m, \delta'). \tag{24}$$

Similarly,

$$\underline{\theta}(n\hat{q}(x); n, \delta') \leq q_K(x) \leq \overline{\theta}(n\hat{q}(x); n, \delta'). \tag{25}$$

From (22), (23), (24), and (25), the following holds:

$$\begin{aligned} \underline{\theta}(m\hat{p}(x); m, \delta') - E &\leq v(x) \cdot p(x) \leq \overline{\theta}(m\hat{p}(x); m, \delta') + E \text{ and} \\ \underline{\theta}(n\hat{q}(x); n, \delta') - E &\leq v(x) \cdot q(x) \leq \overline{\theta}(n\hat{q}(x); n, \delta') + E. \end{aligned}$$

Therefore, for any $x \in B_j$, $w^*(x)$ is bounded as follows:

$$\frac{\underline{\theta}(n\hat{q}(x); n, \delta') - E}{\overline{\theta}(m\hat{p}(x); m, \delta') + E} \leq w^*(x) = \frac{q(x)}{p(x)} \leq \frac{\overline{\theta}(n\hat{q}(x); n, \delta') + E}{\underline{\theta}(m\hat{p}(x); m, \delta') - E}.$$

Since we apply the Clopper-Pearson interval for $K$ partitions for both source and target, the claim holds due to the union bound.

# E ADDITIONAL ALGORITHMS

## E.1 PS ALGORITHM

---
**Algorithm 2** PS: an algorithm using the CP bound in (3)

---
**procedure** $\text{PS}(S_m, f, \mathcal{T}, \varepsilon, \delta)$
    $\hat{\tau} \leftarrow 0$
    **for** $\tau \in \mathcal{T}$ **do**                         ($\triangleright$) Grid search in ascending order
        **if** $U_{\text{CP}}(C_\tau, S_m, \delta) \leq \varepsilon$ **then**
            $\hat{\tau} \leftarrow \max(\hat{\tau}, \tau)$
        **else**
            **break**
    **return** $\hat{\tau}$

---

## E.2 PS-C ALGORITHM

---
**Algorithm 3** PS-C: an algorithm using the CP bound in (3) with $\varepsilon/b$

---
**procedure** $\text{PS-C}(S_m, f, \mathcal{T}, b, \varepsilon, \delta)$
    **return** $\text{PS}(S_m, f, \mathcal{T}, \varepsilon/b, \delta)$

---

We describe the PS-C algorithm, which uses a conservative upper bound on the CP interval. Let

$$L_P(C) \coloneqq \mathop{\mathbb{E}}_{(x,y)\sim P} \left[ \mathbb{1}\left(y \notin C(x)\right) \right]$$

$$L_Q(C) \coloneqq \mathop{\mathbb{E}}_{(x,y)\sim Q} \left[ \mathbb{1}\left(y \notin C(x)\right) \right]$$

$$w^*(x) \coloneqq \frac{q(x)}{p(x)}$$

$$b \coloneqq \max_{x \in \mathcal{X}} w^*(x).$$

Then, we have

$$
\begin{aligned}
L_Q(C) &= \mathop{\mathbb{E}}_{(x,y)\sim Q} \left[ \mathbb{1}\left(y \notin C(x)\right) \right] \\
&= \mathop{\mathbb{E}}_{(x,y)\sim P} \left[ w^*(x)\mathbb{1}\left(y \notin C(x)\right) \right] \\
&\leq \mathop{\mathbb{E}}_{(x,y)\sim P} \left[ b \cdot \mathbb{1}\left(y \notin C(x)\right) \right] \\
&= b \mathop{\mathbb{E}}_{(x,y)\sim P} \left[ \mathbb{1}\left(y \notin C(x)\right) \right] \\
&= b \cdot L_P(C).
\end{aligned}
$$

Thus, $L_Q(C) \leq \varepsilon$ if $b \cdot L_P(C) \leq \varepsilon$. Equivalently, $L_Q(C) \leq \varepsilon$ if $L_P(C) \leq \varepsilon/b$. As a consequence, we can choose $C$ based on the CP bound for the i.i.d. case (i.e., Algorithm 2), except using the desired error of $\varepsilon/b$ (instead of $\varepsilon$). The algorithm is described in Algorithm 3.

### E.3  PS-R ALGORITHM

---

**Algorithm 4** PS-R: an algorithm using the RSCP bound in (6)

---

**procedure** PS-R$(S_m, f, \mathcal{T}, w, b, \varepsilon, \delta)$
    $V \sim \text{Uniform}([0,1])^m$
    $\hat{\tau} \leftarrow 0$
    **for** $\tau \in \mathcal{T}$ **do**                                       ($\triangleright$) Grid search in ascending order
        **if** $U_{\text{RSCP}}(C_\tau, S_m, V, w, b, \delta) \leq \varepsilon$ **then**
            $\hat{\tau} \leftarrow \max(\hat{\tau}, \tau)$
        **else**
            **break**
    **return** $\hat{\tau}$

---

### E.4  PS-M ALGORITHM

---

**Algorithm 5** PS-M: an algorithm using the RSCP bound in (6) along with IWs rescaling

---

**procedure** PS-M$(S_m, T_n^X, f, \mathcal{T}, w, b, \varepsilon, \delta)$
    $\hat{w}(x) \leftarrow \frac{\hat{q}_B(x)}{\hat{p}_B(x)}$ for $x \in \mathcal{X}$, where $\hat{p}_B$ and $\hat{q}_B$ are defined in (14) and (15), respectively
    **return** PS-R$(S_m, f, \mathcal{T}, \hat{w}, b, \varepsilon, \delta)$

---

## F  EXPERIMENT DETAILS

### F.1  DOMAIN ADAPTATION

We use a fully-connected network (with two hidden layers, where each layers has 500 neurons followed by ReLU activations and a 0.5-dropout layer) as the domain classifier (recall that the input of this domain classifier is the last hidden layer of ResNet101). We use the last hidden layer of the model as example space $\mathcal{X}$, where its dimension is 2048. For neural network training, we run stochastic gradient descent (SGD) for 100 epochs with an initial learning rate of 0.1, decaying it by half once every 20 epochs. The domain adaptation regularizer is gradually increased as in (Ganin et al., 2016). We use the same hyperparameters for all experiments.

### F.2  DOMAINNET

We split the dataset into 409,832 training, 88,371 calibration, and 88,372 test images.

### F.3  IMAGENETC-13

We split ImageNet into 1.2M training, 25K calibration, and 25K test images, and ImageNet-C13 into 83M training, 1.6M calibration, and 1.6M test images.

To train a model using domain adaptation, due to the large size of the target training set, we subsample the target training set to be the same size as the source training set on for each random trials.

## G  ADDITIONAL RESULTS

### G.1  SYNTHETIC RATE SHIFT BY TWO GAUSSIANS

We demonstrate the efficacy of the proposed approaches (i.e., PS-R with the known IWs and PS-W with the estimated IWs) using a synthetic dataset consisting of samples from two Gaussian distributions.

**Dataset.** We consider two Gaussian distributions $\mathcal{N}(\mu, \Sigma)$ and $\mathcal{N}(\mu, \Sigma')$ over 2048-dimensional covariate space $\mathcal{X}$. Here, $\mu = \mathbf{0}$; $\Sigma$ and $\Sigma'$ are diagonal where $\Sigma_{1,1} = 5^2$, $\Sigma_{i,i} = 10^{-1}$, $\Sigma'_{1,1} = 1$,

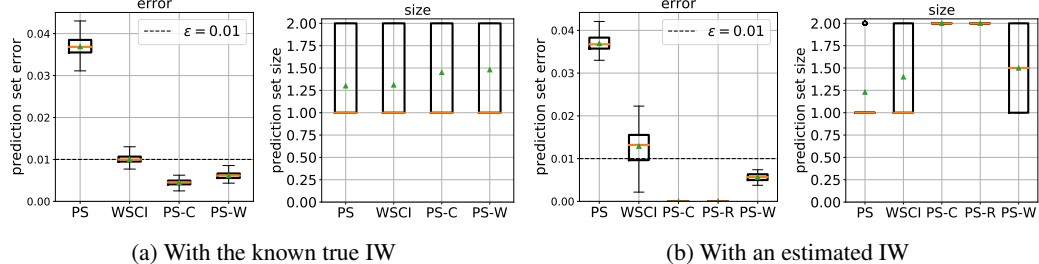

(a) With the known true IW  (b) With an estimated IW

Figure 3: Error under the rate shift by Two Gaussians (over 100 random trials). Parameters are $m = 50,000$, $\varepsilon = 0.01$, and $\delta = 10^{-5}$.

and $\Sigma'_{i,i} = 10^{-1}$ for $i \in \{2, \ldots, 2048\}$. We consider the "flat" Gaussian $\mathcal{N}(\mu, \Sigma)$ as the source and the "tall" Gaussian $\mathcal{N}(\mu, \Sigma')$ as the target. Intuitively, there is a rate shift from the source to the target—i.e., the target examples are a subset of the source, but occur with higher frequencies. We use the following labeling function: $p(y \mid x) = \sigma(5x_1)$, where $\sigma$ is the sigmoid function. Finally, we generate 50,000 labeled examples for each training, calibration, and test.

**Results.** We consider two different setups: 1) the true IW is known, and 2) the true IW is unknown. In Figure 3a, we demonstrate the prediction set errors given the true IW. As expected PS-R satisfies the PAC guarantee—i.e., the error is below $\varepsilon$. However, as shown in Figure 3b, when we need to estimate IWs, using just the point-estimate of the IW results in PS-R performing poorly in terms of prediction set error; it still satisfies the $\varepsilon$ constraint, but the error is close to zero, indicating that the prediction set size is too large to be useful for an uncertainty quantifier. In contrast, PS-W (i.e., rejection sampling based on interval estimates of IWs) produces a larger, more reasonable error rate while still satisfying the PAC condition. These experiments demonstrate that PS-R works well when given the true IW, but accounting for IW uncertainty is important when using estimated IWs.

## G.2 PREDICTION SET SIZE AND ERROR

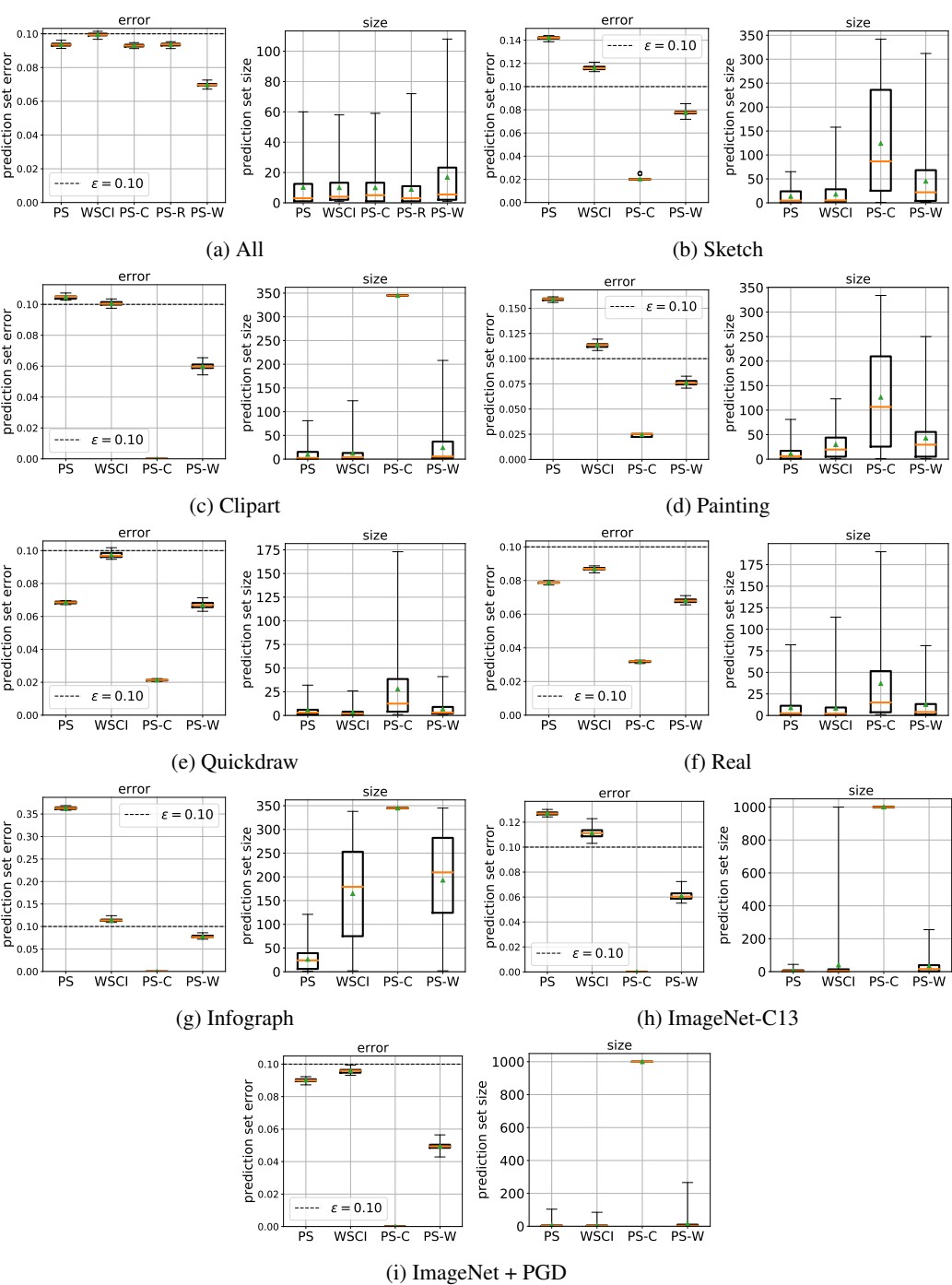

Figure 4: Prediction set error and size under rate shifts on DomainNet (a-g) and under support shifts on ImageNet (h, i) (over 100 random trials for error and over a held-out test set for size). Parameters are $m = 50,000$ for DomainNet and $m = 20,000$ for ImageNet, $\varepsilon = 0.1$, and $\delta = 10^{-5}$.

### G.3 ABLATION STUDY ON CALIBRATION

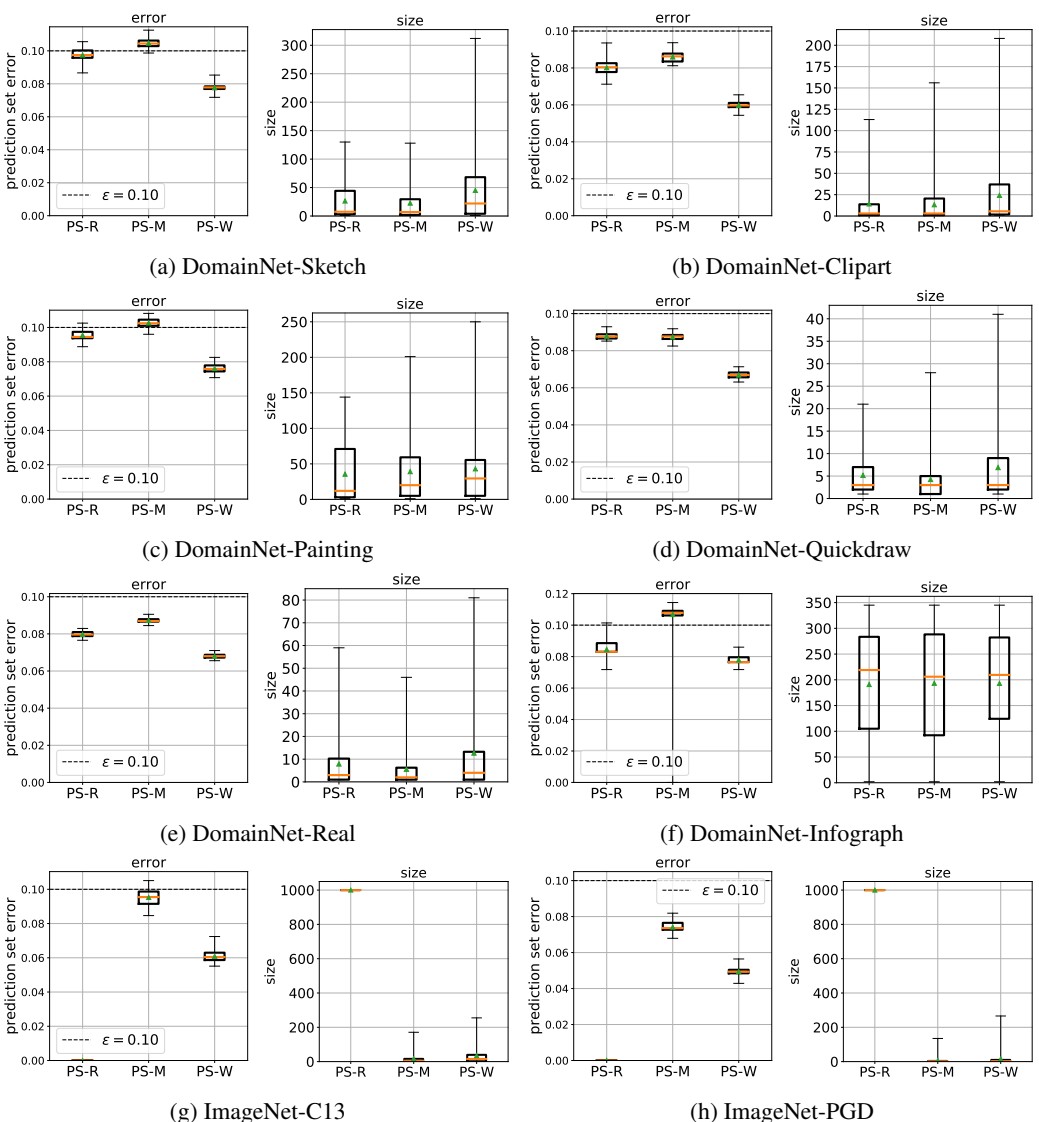

Figure 5: Ablation study on calibration (over 100 random trial for error and over a held-out test set for size). PS-R does not use rescaled IWs, PS-M uses point estimates of rescaled IWs, and PS-W uses intervals around rescaled IWs. Parameters are $m = 50,000$ for DomainNet shifts, $m = 20,000$ for ImageNet shifts, $\varepsilon = 0.1$, and $\delta = 10^{-5}$. In particular, we consider a variant PS-M of PS-W that ignores the worst-case IWs as in Theorem 3; instead, it rescales the importance weights in each bin via a point estimate, i.e., Theorem 5 without the Clopper-Pearson interval (i.e., $m = n = \infty$) and with $E = 0$. As can be seen in the results on the shift to various domains, our approach satisfies the PAC guarantee but this version does not—in fact, its error is even worse than PS-R, which uses the non-rescaled importance weights from the probabilistic classifier $s$.

## G.4 COMPARISON WITH VARIOUS TOP-$K$ PREDICTION SETS

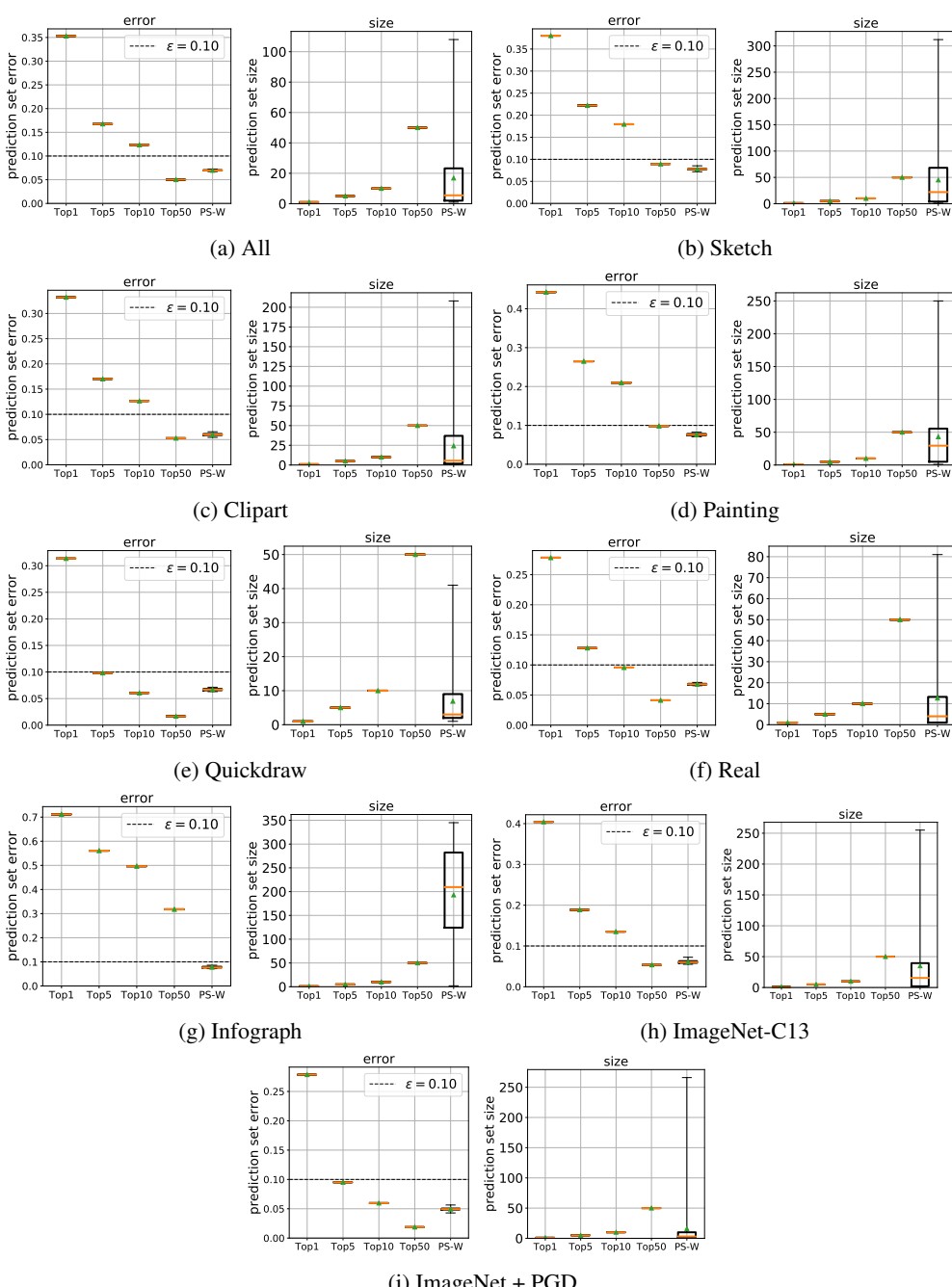

Figure 6: Prediction set error size size under rate shifts on DomainNet (a-g) and under support shifts on ImageNet (h, i) (over 100 random trial). Default parameters are $m = 50,000$ for DomainNet and $m = 20,000$ for ImageNet, $\varepsilon = 0.1$, and $\delta = 10^{-5}$. A Top-$K$ prediction set is a prediction set that contains top $K$ labels based on a domain-adapted score function; thus the set size is always $K$. As can be seen in the prediction set error plots, Top-$K$ prediction sets do not consistently satisfy the desired $\epsilon$ guarantee. Moreover, the prediction set size is worse than PS-W. For example, when the Top-50 prediction set error rate almost achieves the desired error rate, e.g., (6d), the mean and medial of the corresponding prediction set size is larger than PS-W.

## G.5 Varying a Smoothness Parameter

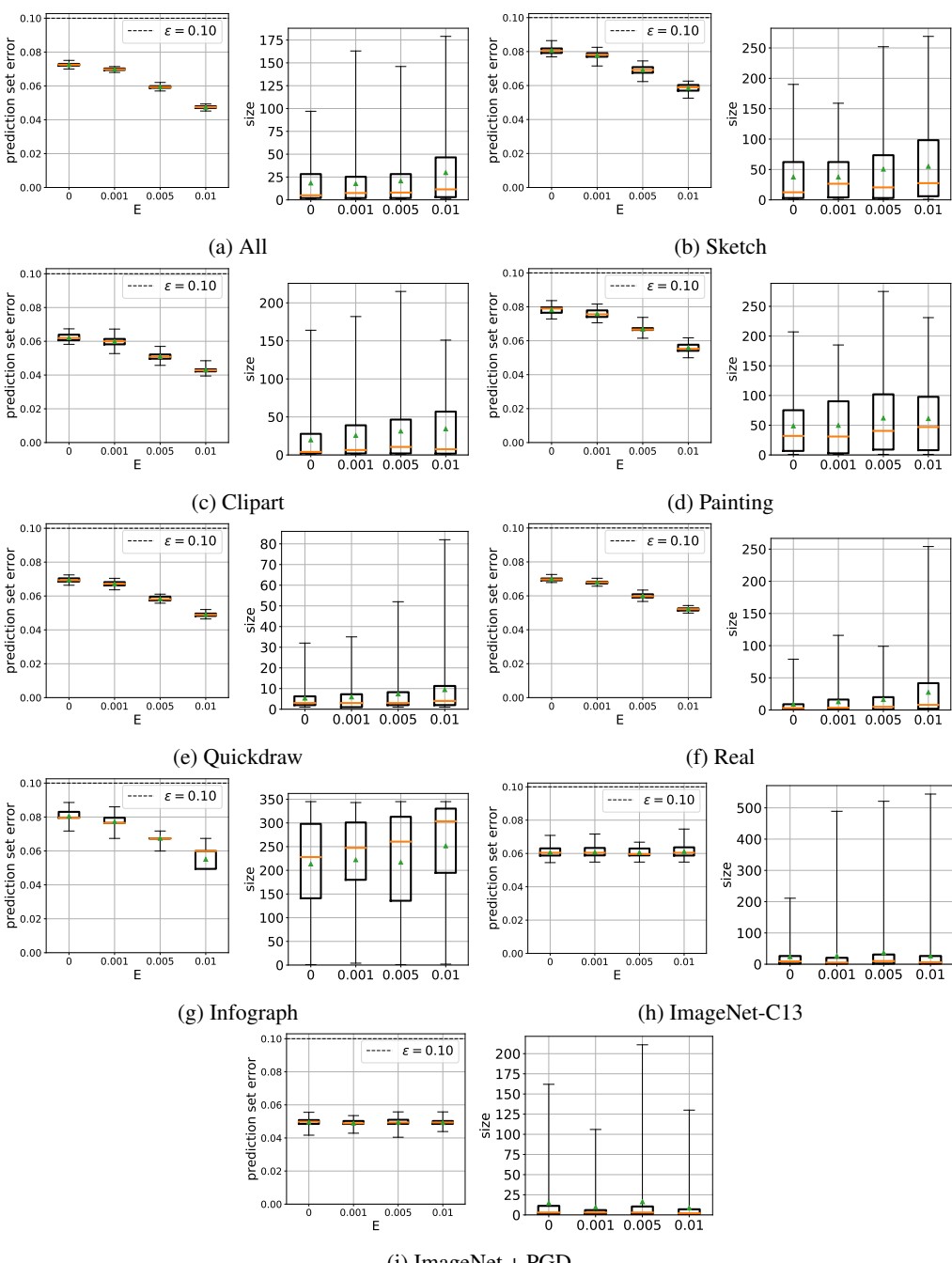

Figure 7: Prediction set error size size under rate shifts on DomainNet (a-g) and under support shifts on ImageNet (h, i) (over 100 random trial) for varying $E$. Parameters are $m = 50,000$ for DomainNet and $m = 20,000$ for ImageNet, $\varepsilon = 0.1$, and $\delta = 10^{-5}$. In particular, the parameter $E$ in Assumption 1 bounds the quality of our estimates of $p(x)$ and $q(x)$; since these errors cannot be conveniently measured, we have chosen it heuristically as a hyperparameter. In this figure, we show the error of PS-W as a function of $E$. As $E$ becomes smaller, the prediction sets become more optimistic while still satisfying the PAC guarantee. Note that the optimal case is $E = 0$, since PS-W still satisfies the PAC guarantee; this result suggests that our IW estimates are reasonably accurate.

## G.6 VARYING A NUMBER OF BINS

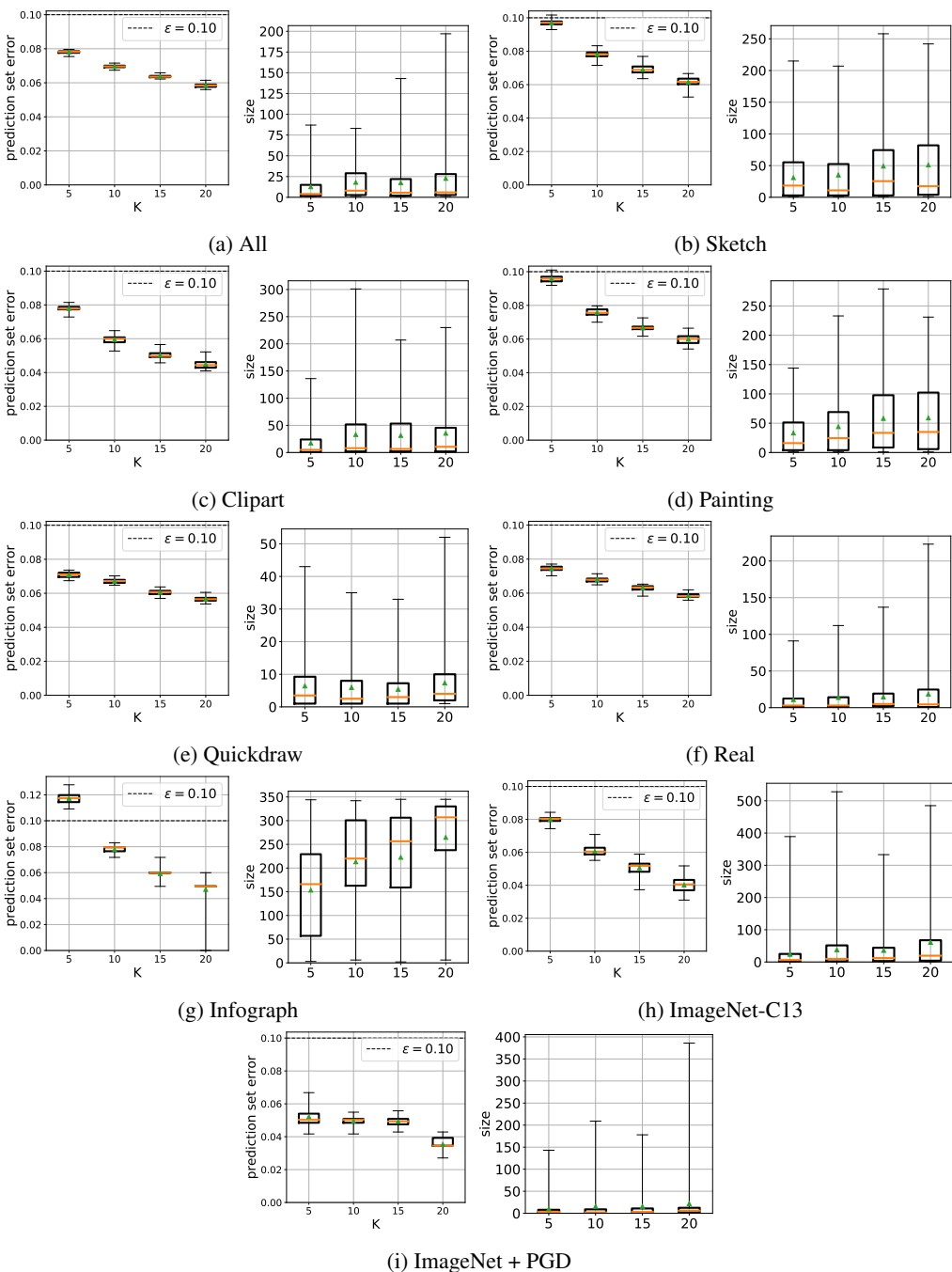

Figure 8: Prediction set error size size under rate shifts on DomainNet (a-g) and under support shifts on ImageNet (h, i) (over 100 random trial) for varying $K$. Parameters are $m = 50,000$ for DomainNet and $m = 20,000$ for ImageNet, $\varepsilon = 0.1$, and $\delta = 10^{-5}$. In general, $K$ must be chosen to be small enough so each bin contains sufficiently many source examples to achieve a small Clopper-Perason interval size (e.g., $10^{-3}$), though it also needs to be sufficiently large to satisfy the smoothness assumption.

## G.7 PREDICTION SET VISUALIZATION

Figure 9: Prediction sets of the DomainNet shift from All to Paint. Parameters are $m = 50,000$, $\varepsilon = 0.1$, and $\delta = 10^{-5}$. The green label is the true label and the label with the hat is the predicted label. We choose examples where the two approaches differ; in particular, if PS-W is incorrect, then PS is incorrect as well since the prediction set sizes are monotone in $\tau$.

Figure 10: Prediction sets of the shift from ImageNet to ImageNet-C13. Parameters are $m = 20,000$, $\varepsilon = 0.1$, and $\delta = 10^{-5}$. The green label is the true label and the label with the hat is the predicted label. We choose examples where the two approaches differ; in particular, if PS-W is incorrect, then PS is incorrect as well since the prediction set sizes are monotone in $\tau$.

| Example $x$ | $\hat{C}_{\text{PS}}(x)$ | $\hat{C}_{\text{PS-W}}(x)$ | Example $x$ | $\hat{C}_{\text{PS}}(x)$ | $\hat{C}_{\text{PS-W}}(x)$ |
|---|---|---|---|---|---|
|  | $\{\widehat{\text{box turtle}}\}$ | $\{\text{terrapin}, \widehat{\text{box turtle}}\}$ |  | $\{\text{brain coral}, \widehat{\text{starfish}}, \text{sea urchin}\}$ | $\{\text{brain coral}, \text{chiton}, \widehat{\text{starfish}}, \text{sea urchin}, \text{sea cucumber}, \text{coral reef}, \text{stinkhorn}\}$ |
|  | $\{\widehat{\text{white terrier}}, \text{kuvasz}, \text{komondor}\}$ | $\{\text{Maltese dog}, \widehat{\text{white terrier}}, \text{kuvasz}, \text{komondor}, \text{Samoyed}\}$ |  | $\{\widehat{\text{kit fox}}\}$ | $\{\text{red fox}, \widehat{\text{kit fox}}\}$ |
|  | $\{\widehat{\text{ladybug}}\}$ | $\{\text{leaf beetle}, \widehat{\text{ladybug}}\}$ |  | $\{\widehat{\text{tusker}}\}$ | $\{\widehat{\text{tusker}}, \text{Afri. elephant}\}$ |
|  | $\{\text{banjo}, \text{elec. guitar}, \widehat{\text{stage}}\}$ | $\{\text{aco. guitar}, \text{banjo}, \text{elec. guitar}, \widehat{\text{stage}}\}$ |  | $\{\text{beaker}, \widehat{\text{pop bottle}}, \text{water bottle}, \text{wine bottle}\}$ | $\{\text{beaker}, \text{beer bottle}, \text{perfume}, \widehat{\text{pop bottle}}, \text{water bottle}, \text{wine bottle}\}$ |
|  | $\{\widehat{\text{cuirass}}\}$ | $\{\text{breastplate}, \widehat{\text{cuirass}}\}$ |  | $\{\text{convertible}, \widehat{\text{sports car}}\}$ | $\{\text{car wheel}, \text{convertible}, \widehat{\text{sports car}}\}$ |
|  | $\{\widehat{\text{confectionery}}\}$ | $\{\widehat{\text{confectionery}}\}$ |  | $\{\widehat{\text{crash helmet}}\}$ | $\{\text{bonnet}, \widehat{\text{crash helmet}}, \text{football helmet}\}$ |

Figure 11: Prediction sets of the shift from ImageNet to ImageNet-PGD. Parameters are $m = 20,000$, $\varepsilon = 0.1$, and $\delta = 10^{-5}$. The green label is the true label and the label with the hat is the predicted label. We choose examples where the two approaches differ; in particular, if PS-W is incorrect, then PS is incorrect as well since the prediction set sizes are monotone in $\tau$.

