# OpenReview forum: "PAC Prediction Sets Under Covariate Shift"
_ICLR.cc/2022/Conference — ICLR 2022 Poster_

### Official Review · Reviewer_ZLVy · 2021-10-28

**Correctness:** 3
**Technical Novelty And Significance:** 3
**Empirical Novelty And Significance:** 3
**Recommendation:** 6
**Confidence:** 3

**Details Of Ethics Concerns:**

I don't have any ethical concerns regarding this paper.

**Main Review:**

Strengths:
1.	The paper proposes a nice solution to a popular and important problem—uncertainty quantification for machine learning models in the presence of covariate shifts;
2.	The proposed method and PAC theory are general to quantify uncertainty for any black-box machine learning models;
3.	The datasets used to evaluate the algorithm are well connected with applications.
4.	The paper is well organized and easy to follow.

Weaknesses and Suggestions:

As shown in Algorithm 1, the random variable V is sampled once. The algorithm is essentially a randomized algorithm. If the upper bound for importance weight b is large, the acceptance rate in rejection sampling can be small, i.e., the random set $T_N$ in Equation (4) can be small. I concern this will lead to some instability issue of the algorithm. In Theorem 2, the high-confidence bound is marginal overall all possible V. This means that conditional on some value of V, the prediction sets given by the algorithm can possibly fail to achieve the target coverage rate.

Section 3.3 needs to clarify the assumptions in Appendices.

1. The proof of Theorem 5 in Appendix D.6 uses Equation (11) in Assumption 1. The authors need to explain how to construct a partition that satisfies the Assumption 1 on page 14. Like the authors said in Appendix B.2, In general, we need exponentially many bins to guarantee Assumption 1. I encourage the authors to add a formal proposition for the uncertainty set which states clearly the assumption needed to achieve the high-probability bound in Equation (6). It is okay if this assumption can only deal with by some heuristic approach. But it is important to explain what you did in practice, e.g., Why it is a good choice to estimate the lower and upper bound of the true IWs with $E=0.01$ and $K=10$ bins in Section 4.1?

2. Appendix C.1 states that the uncertainty set defined below Equation (6) does not consider the fact that the expected importance weight is one; the optimization problem with additional constraint is challenging to solve exactly. The authors provide a good intuition of the greed algorithm using the monotonicity of $U_{RSCP}$. However, if the lower and upper bound of the true IWs are estimated, the constraint in Equation (10) is not the same as the constraint in Equation (7), so I don’t see why Theorem 4 holds by Equation (9) and using the same argument as the proof of Theorem 3. Also, in footnote 5, the authors mention “we assume b is given for simplicity, but our approach estimates it;” This also leads to some difficulty for moving from Theorem 3 to 4 as well. At the end of Appendix A, the authors mentioned “Compared to Candes et al. (2021), which only guarantees asymptotic validity under certain assumptions on the estimated IWs, Theorem 4 provides a significantly stronger finite-sample correctness guarantee (in addition to providing PAC-style guarantees).” However, I am concerned if Theorem 4 is only true if we can find a partition that achieves Assumption 1 and we know the true maximum importance weight $b$.

In experiments, the benchmark WSCI (Tibshirani et al., 2019) approximately achieves the target error rate $0.1$ and often have smaller prediction sets than PS-W (Figure 2). The experiments on real-world datasets are truly interesting. However, it may be better to compare WSCI and PS-W on a simpler synthetic dataset assuming the true importance weights are given.

1. I wonder if your method based on PAC theory can provide smaller prediction intervals than the conformal prediction approach WSCI. Theoretically, could we have a comparison between WSCI and PS-W assuming the true importance weights are known?

2. WSCI estimates the importance weights. Your method does rejection sampling using the importance weights, it would be great if the authors can add a discussion comparing both approaches.

3. WSCI only assumes the weighted exchangeability of the data. Your method assumes the data are i.i.d, which is a stronger assumption and need to be mentioned clearly in the paper.



**Summary Of The Paper:**

This paper proposes a new method to construct approximately correct (PAC) prediction sets for uncertainty quantification in the presence of covariate shift. It is a natural and interesting extension of the previous works [Park et al. 2020a, 2021] on PAC prediction sets. The building blocks this paper took from these previous works are the optimization problem in Equation (1) and the Clopper-Pearson confidence intervals for the Binomial distribution in Section 2.2. The extension is based on a rejection sampling Clopper-Pearson bound given in Section 3.2.
The authors propose an algorithm with and without access to the true important weights. The algorithm is evaluated in the settings of “rate shift” and “support shift” on the DomainNet and ImageNet datasets. The experiments show that the algorithm gives the smallest prediction sets among approaches that always satisfy the PAC constraint.


**Summary Of The Review:**

The method proposed by the paper is interesting and new. However, I found some claims of the theories in the paper need to be clarified or made more formally. I hope the authors can address my concerns in their feedback.

---

> ### Author Response · Authors · 2021-11-23
> **Author Response (1/2)**
>
> Thanks for your valuable comments and suggestions; we have responded below and updated our paper accordingly. We also highlighted the updated parts in red.
>
> ---
> > As shown in Algorithm 1, the random variable V is sampled once. The algorithm is essentially a randomized algorithm. If the upper bound for importance weight b is large, the acceptance rate in rejection sampling can be small, i.e., the random set T_N in Equation (4) can be small. I concern this will lead to some instability issue of the algorithm. In Theorem 2, the high-confidence bound is marginal overall all possible V. This means that conditional on some value of V, the prediction sets given by the algorithm can possibly fail to achieve the target coverage rate.
>
> The overall failure rate (including the randomness of $V$) is bounded by $\delta$, which can be taken to be very small. If desired, we can use multiple $V$, compute the prediction set for each, and then intersect the prediction sets, as long as we union bound over \delta (i.e., if we take $K$ samples $V$, then we get ($\epsilon$, $K*\delta$) PAC prediction sets. The reason is that *each* of the $K$ prediction sets is valid with probability $1-\delta$. We did not find a need to do so in practice.
>
> ---
> > The proof of Theorem 5 in Appendix D.6 uses Equation (11) in Assumption 1. The authors need to explain how to construct a partition that satisfies the Assumption 1 on page 14. Like the authors said in Appendix B.2, In general, we need exponentially many bins to guarantee Assumption 1. I encourage the authors to add a formal proposition for the uncertainty set which states clearly the assumption needed to achieve the high-probability bound in Equation (6). It is okay if this assumption can only deal with by some heuristic approach.
>
> For Theorems 3 & Theorem 4, we assume that IW uncertainty sets satisfying the high-probability bound in Equation (6) are given (i.e., Theorems 3 & 4 hold assuming this condition is satisfied). We believe that there can be multiple ways to find the uncertainty set with this property, but we introduce one way to estimate this uncertainty set as described in Appendix B.
>
> For constructing a partition that satisfies Assumption 1, if the source PDF $p(x)$ and target PDF $q(x)$ are $L$ Lipschitz continuous, then each bin $B_j$ can be constructed such that $\|x - x'\| \le E / L / Volume(B_j)$ for any $x, x' \in B_j$; the bins constructed in this way are guaranteed to satisfy Assumption 1. See Appendix B.3 for full details.
>
> ---
> > But it is important to explain what you did in practice, e.g., Why it is a good choice to estimate the lower and upper bound of the true IWs with E=0.01 and K=10 bins in Section 4.1?
>
> Choosing hyperparameters E and K is non-trivia and we use the following heuristics that expect to give us a satisfactory estimate of the lower and upper bound of the true IWs.
>
> *On the choice of $K$.* Our bin is defined in one dimensional space (as mentioned in Appendix B.2), so we follow the usual practice in the calibration literature with respect to binning (e.g., Guo et al., 2017 or Park et al., 2021), where $K$ is about 10 to 20. As we are using equal-mass binning, we choose the number of bins so each bin contains sufficiently many source samples (e.g., 5000 samples) for the Clopper-Pearson interval over IWs for each bin is small (e.g., around 1e-3 for various confidence levels); this leads us to $K=10$. We also added a sensitivity analysis for $K$ for all shifts to Appendix G.6.
>
> *On the choice of E.* We use $E=0.001$ in all experiments; we updated this typo. We believe that estimating $E$ (i.e., computing the integral of the difference of source and target probabilities) is intractable since it requires us to perform density estimation in a high-dimensional space (i.e., 2048 in our case), and then compute an integral over each bin. In practice, we find that we can simply choose $E=0$ (or close to zero, e.g., 0.001). Intuitively, we find that this choice works well since binning by the source-discriminator score $g(x)$ is an effective way to group points with similar IWs; rescaling the IWs is only necessary since $g(x)$ might be overconfident, leading to inflated IWs. In Appendix G.5, we provide a sensitivity analysis for $E$ for all of our shifts. Importantly, note that PS-W with $E=0$ satisfies PAC criterion in all cases.
>
> We have added our discussion to Appendix C.3.

---

> ### Author Response · Authors · 2021-11-23
> **Author Response (2/2)**
>
> >Appendix C.1 states that the uncertainty set defined below Equation (6) does not consider the fact that the expected importance weight is one; the optimization problem with additional constraint is challenging to solve exactly. The authors provide a good intuition of the greed algorithm using the monotonicity of U_{RSCP}. However, if the lower and upper bound of the true IWs are estimated, the constraint in Equation (10) is not the same as the constraint in Equation (7), so I don’t see why Theorem 4 holds by Equation (9) and using the same argument as the proof of Theorem 3.
>
> Theorem 4 is not for estimated lower/upper bounds; it assumes given correct lower/upper bounds (i.e., they contain the true IWs). In particular, Theorems 3 & 4 both assume that the IWs satisfy Eq. 6, which says that the true IW is contained in the given interval (the estimation error $\delta_w$ is for simplicity in stating subsequent results). As a consequence, Theorem 4 holds due to Theorem 3 and (9). We have clarified this point in the statements of Theorems 3 & 4.
>
> ---
> >Also, in footnote 5, the authors mention “we assume b is given for simplicity, but our approach estimates it;” This also leads to some difficulty for moving from Theorem 3 to 4 as well.
>
> Under Assumption 1, we can rigorously estimate $b$ and use it with Theorem 4; we have added details in Appendix C.2 (in particular, see Theorem 6). This is unrelated to Theorem 4, since Theorems 3 & 4 both assume that a true upper bound b is given.
>
> ---
> > At the end of Appendix A, the authors mentioned “Compared to Candes et al. (2021), which only guarantees asymptotic validity under certain assumptions on the estimated IWs, Theorem 4 provides a significantly stronger finite-sample correctness guarantee (in addition to providing PAC-style guarantees).” However, I am concerned if Theorem 4 is only true if we can find a partition that achieves Assumption 1 and we know the true maximum importance weight b.
>
> First, we note that Candes et al. (2021) makes similar assumptions; in particular, Theorem 4 in their appendix (pg. 26) makes the assumption that the IWs are bounded (Assumption (3) in their theorem statement), and furthermore that the IW estimates converge asymptotically (Assumption (4) in their theorem statement). Similarly, our Theorem 4 is true under the bounded IW assumption and that the true IW is contained in the IW intervals (the latter assumption is achieved if Assumption 1 holds, but other strategies might be possible).
> Making similar assumptions is necessary, since in general estimating IWs is impossible without using a nonparametric density estimator (even so, we would need assumptions such as that the densities p and q are Lipschitz continuous; pathological densities are impossible to handled rigorously).
> Finally, we note that the maximum IW assumption can be removed under Assumption 1; we have added details to Appendix C.2 (in particular, Theorem 6).
>
> ---
> > I wonder if your method based on PAC theory can provide smaller prediction intervals than the conformal prediction approach WSCI. Theoretically, could we have a comparison between WSCI and PS-W assuming the true importance weights are known?
>
> The main difference between WSCI and PS-W lies in the guarantee provided (rather than prediction set size); PS-W provides a stronger guarantee than WSCI. In particular, WSCI does not provide a PAC guarantee. Instead, to satisfy the coverage probability guarantee, it requires a new calibration set for every new test example. However, in practice, we usually have a single held-out calibration set. Thus, our approach PS-W provides a guarantee that holds conditioned on this set. This difference is illustrated in Figure 3(a), which compares WSCI and PS-W given the true importance weight. PS-W produces a larger set size than WSCI, but strictly satisfies the error constraint. The shortcoming of WSCI can also be observed in Figure 2 in Tibshirani et al. (2019), which empirically shows that the guarantee only holds on average over examples. We have added this discussion to Appendix C.4.
>
> ---
> > WSCI estimates the importance weights. Your method does rejection sampling using the importance weights, it would be great if the authors can add a discussion comparing both approaches.
>
> The main difference is that for their theory, WSCI assumes given the true IWs, whereas our method provides theoretical guarantees even with approximate IWs (as long as confidence intervals are given). They also use IWs differently. Given the true importance weights, WSCI uses the importance weights to reweight examples in the calibration set, and PS-W uses the importance weights to generate a target labeled calibration set using rejection sampling; this is added to Appendix C.4.
>
> ---
> >WSCI only assumes the weighted exchangeability of the data. Your method assumes the data are i.i.d, which is a stronger assumption and need to be mentioned clearly in the paper.
>
> We clearly mentioned the weaker assumption of WSCI.

---

### Official Review · Reviewer_vZMq · 2021-11-02

**Correctness:** 4
**Technical Novelty And Significance:** 2
**Empirical Novelty And Significance:** 3
**Recommendation:** 6
**Confidence:** 2

**Main Review:**

Studying prediction sets under covariate shift in the PAC setting is a well-motivated problem given the ubiquitous presence of probabilistic classifiers in machine learning practice. Overall this work is well-written, and does a good job motivating and placing itself within the prior literature.

On the theoretical side, the work is incremental. It relies mostly on previous ideas from the literature in combination with classical techniques like rejection sampling. Further, the authors provide no theoretical guarantee on the optimality of the output set size. This is the main parameter of interesting the prediction set model, since it is easy to get standard PAC guarantees simply by outputting the entire label space. On the other hand, the authors do provide ample empirical evidence that their algorithm returns small prediction sets compared to other baseline models with similar accuracy guarantees.

Typos: "assume given" (bottom of page 4/5).

**Summary Of The Paper:**

Let $X$ be an instance space, $Y$ a set of labels, $D$ some underlying (hidden) distribution over $X \times Y$. This work studies a new method for converting the output of a probabilistic predictor (e.g. a deep net) to a good prediction set: that is a mapping from $C: X \to 2^Y$, such that for most samples $(x,y) \sim D$, $C(x)$ contains $y$. Formally, the authors study this problem in the PAC setting with covariate shift. The learner is given access to the score function, a labeled sample from the source distribution $P$ over $X \times Y$, and unlabeled samples from a target distribution $Q$ over $X \times Y$ whose marginal over $X$ may be shifted from the source. The goal is to output a prediction set which is as small as possible while still retaining PAC guarantees with respect to the shifted target distribution. This is a well-motivated model in practice. Probabilistic outputs of modern neural nets can be difficult to interpret, and small prediction sets may be useful in settings where one wishes to avoid or check more carefully a few marked outputs (e.g. a problematic medical diagnosis).

Assuming known, bounded importance weights, the authors provide an algorithm satisfying the PAC guarantee by maximizing a cutoff value for the score function that performs well over an empirical sample. They ensure their guarantee holds over the target rather than source distribution by rejection sampling to simulate the target distribution. The authors also extend this to settings where the importance weights are unknown but can be estimated from unlabeled samples. Finally, the authors provide experimental evidence over a couple common settings of covariate shift. They show that their algorithm outperforms baseline methods in the literature in the sense that it maintains PAC guarantees while outputting a smaller prediction set  in expectation.


**Summary Of The Review:**

Overall, the submission is borderline. The proposed algorithm is mostly based on prior work and has few non-trivial theoretical guarantees, but it does perform well against other baselines in interesting practical settings such as rate shift.


EDIT: I have read the author response and my review remains largely unchanged, though I would raise the score to 7 if this were possible within the ICLR format. Rejection sampling is a well-known strategy within the domain adaptation community for dealing with bounded covariate shift. The modification to approximate importance weights is more novel, but not of great theoretical interest since it is basically an immediate application of monotonicity. Finally, while the authors do provide a theoretical guarantee on the threshold cutoff parameter $\tau$, this is only meaningful with respect to a very narrow set of algorithms (namely those w/ one fixed cutoff parameter that satisfy PAC-guarantees). There is no obvious way to actually connect this to a guarantee on the optimal prediction set sizes, even in the PAC setting (nor is this ever discussed). However, the empirical results do a good job of making up for these weaknesses, so I would give the paper a 7 if I could. Unfortunately ICLR does not give provide this option.

---

> ### Author Response · Authors · 2021-11-23
> **Author Response**
>
> Thanks for your valuable comments and suggestions; we have responded below and updated our paper accordingly. We also highlighted the updated parts in red.
>
> ---
> > On the theoretical side, the work is incremental. It relies mostly on previous ideas from the literature in combination with classical techniques like rejection sampling.
>
> Our main theoretical contribution is Section 3.3, “Approximate Importance Weights”. To the best of our knowledge, the techniques that we propose for solving the robust prediction set problem are novel. In particular, Theorem 3 encodes a novel strategy for constructing prediction sets with robust IWs, and Lemma 1 and Theorem 4 convert this result into a practical algorithm for PAC prediction sets with approximate IWs. Finally, while rejection sampling is a classical technique, to the best of our knowledge, it has not been previously used for handling covariate shift in machine learning.
>
> ---
> > Further, the authors provide no theoretical guarantee on the optimality of the output set size. This is the main parameter of interesting the prediction set model, since it is easy to get standard PAC guarantees simply by outputting the entire label space. On the other hand, the authors do provide ample empirical evidence that their algorithm returns small prediction sets compared to other baseline models with similar accuracy guarantees.
>
> We note that we are providing theoretical guarantees for our specific algorithm, which empirically produces small prediction sets. In particular, our algorithm maximizes the parameter \tau subject to constraint encoding the desired PAC guarantee, thereby ensuring it does not construct a trivial prediction set. In general, if the scoring function is poor, then outputting the entire label space is the only valid solution, though optimality guarantees might be possible in terms of the quality of the scoring function. However, this direction is more related to existing work on prediction sets in the i.i.d. setting (which also do not provide such guarantees), and is orthogonal to our goals (which is to adapt these techniques to the covariate shift setting).

---

### Official Review · Reviewer_vXNT · 2021-11-04

**Correctness:** 3
**Technical Novelty And Significance:** 3
**Empirical Novelty And Significance:** 2
**Recommendation:** 6
**Confidence:** 3

**Main Review:**

Strengths:
1. Technical details/proofs have been adequately presented

Critical Comments:
1. It would be helpful to include reason why probabilistic classification models that can handle covariate shift not be used for PAC prediction sets. For eg. why not include labels with top-k posterior likelihoods to create the prediction set? etc. In the same spirit, it seems necessary to atleast empirically compare against such baselines.
2. While robust variant in sec3.3 is ok, because the goal is to optimize (guard against) the worst-case scenario, one would expect that the variant would turn out to be more conservative/pessimistic. However, the simulations show that the size of predictions is small than it's non-robust (vanilla) counterpart. Am I missing something?
3. Since the work is about PAC guarantees etc., it seems to be more efficient if the probabilities (uncertainties) in the confidence interval of the importance weights are taken into account and propagated through. However, currently the Confidence intervals are merely approximated as an interval. Can this be improved?
4. Simulations seem to be a bit weak without comparison to top-k methods etc.


Minor Comment:
1. Near footnote 1, "p(x,y) is pdf" ... if y is dicrete like in classification, there is no such pdf.

**Summary Of The Paper:**

The paper presents algorithms for PAC prediction sets under the assumption of covariate shift. Using estimated/known importance weights that encode the shift, the method optimizes for the smallest subset of labels such that with high probability the error of this prediction set is low. A rejection sampling based strategy is then shown to satisfy the PAC constraints. The methodology is extended to the case where the importance weights are uncertain (due to estimation error). It is shown that the robust variant can be solved using the extreme case weights and the rejection sampling based algorithm. Simulations show the efficacy of the method.

**Summary Of The Review:**

While the work seems technically sound, there are some loose ends like those mentioned above. Empirical comparisons with some baselines also seem to be missing. So I recommend only a marginal accept in it's current state.

---

> ### Author Response · Authors · 2021-11-23
> **Author Response**
>
> Thanks for your valuable comments and suggestions; we have responded below and updated our paper accordingly. We also highlighted the updated parts in red.
>
> ---
> > It would be helpful to include reason why probabilistic classification models that can handle covariate shift not be used for PAC prediction sets. For eg. why not include labels with top-k posterior likelihoods to create the prediction set? etc. In the same spirit, it seems necessary to at least empirically compare against such baselines.
>
> Such models can in fact be used for PAC prediction sets; currently, our score function is based on the domain adapted classifier. See the answer below as well.
>
> ---
> > Simulations seem to be a bit weak without comparison to top-k methods etc.
>
> First, we note that the top-K prediction set does not provide theoretical guarantees, whereas our approach does (in particular, it provides a PAC guarantee). Nevertheless, we have added the top-K prediction sets results with a domain adapted score function (where k=1, 5, 10, 50) to Appendix G.4. This baseline produces a prediction set with a constant set size. Note that if K is sufficiently large, then the prediction sets satisfy the PAC guarantee; however, in these cases, the median set size is larger than those constructed using our approach. On the other hand, if K is small, then the set size is small, but frequently violates the PAC guarantee.
>
> Furthermore, additional theory is needed to choose the appropriate value of K in a rigorous way, which could be especially difficult without labeled target data for cross-validation. Thus, our new results demonstrate that our proposed approach is strictly better than top-K prediction sets, both theoretically and practically.
>
> ---
> > While robust variant in sec3.3 is ok, because the goal is to optimize (guard against) the worst-case scenario, one would expect that the variant would turn out to be more conservative/pessimistic. However, the simulations show that the size of predictions is small than it's non-robust (vanilla) counterpart. Am I missing something?
>
> Our ablation is non-robust, and also does not use our IW calibration strategy, which produces significantly better calibrated IWs. Thus, the poorly calibrated weights can be either overly conservative or overly optimistic. We show an ablation that is non-robust but uses calibration in Figure 5 in Appendix G.3; as can be seen, the prediction sets are always smaller, but more likely to be overconfident and exceed the desired error bound.
>
> ---
> >Since the work is about PAC guarantees etc., it seems to be more efficient if the probabilities (uncertainties) in the confidence interval of the importance weights are taken into account and propagated through. However, currently the Confidence intervals are merely approximated as an interval. Can this be improved?
>
> While we could compute a distribution over IWs, this object is significantly more high-dimensional and therefore harder to estimate correctly. In contrast, confidence intervals consist of just two values, making it more plausible that our intervals correctly capture the true uncertainty. Nevertheless, we will note this as a possible direction for future work.
>
> ---
> > Near footnote 1, "p(x,y) is pdf" ... if y is dicrete like in classification, there is no such pdf.
>
> We refer to footnote 1 for an explanation of why we can use densities here. We have clarified this further. For a classification problem, we define the source distribution $P$ to be a probability measure on $X \times Y$, defined with respect to a base measure $M$ on $X \times Y$, which is a product of the Lebesgue measure on $X$ (assuming $X$ is a measurable subset of $R^d$ for some $d>0$, which is always the case for us) and the counting measure on $Y$.  Then, the density is defined to be the Radon-Nykodym derivative $p = dP/dM$ of $P$ with respect to $M$.

---

### Official Review · Reviewer_qXST · 2021-11-09

**Correctness:** 3
**Technical Novelty And Significance:** 3
**Empirical Novelty And Significance:** 4
**Recommendation:** 8
**Confidence:** 4

**Main Review:**

Strengths:

I think the paper studies an interesting approach to a relevant problem. It is cleanly written, and the method is developed slowly and clearly, at least up to the final piece of estimating the $\mathcal{W}$, but this final opacity is fine since this part largely uses established methods. In particular I think the idea of using the sample-wise bounds on $w_i$ to come up with worst-case type bounds is quite clever. The presented method shows meaningful gains over both prior proposals (WSCI), and the study is ablated well to illustrate the effects of each of the pieces of the main scheme.

Weaknesses:

To me there are two main weaknesses:

- Firstly, no real justification for the choice of $E = 0.01$ and $K = 10$ is presented. How were these numbers arrived at? Is it reasonable that such crude chunking of the data yields such regular marginals? For that matter, no contextualisation of these numbers is given either - how optimistic or not is assuming $E = 0.01$? Finally, sensitivity with respect to these hyperparameters is not discussed. I think understanding these aspects is critical to the practical usage of a method like this.

- Secondly, I don't think the conclusion is as clear as is suggested by the "mean-normalised-size-of-reliable-methods" metric. This mean normalised size itself is not too meaningful as a metric to me (these are fairly different tasks with fairly different scales of difficulty as evinced by the mean sizes themselves). More importantly, I find it a much more natural statement to say that PS-R as a method makes sense when the shift is non-adversarial, and PS-M as a method that is slightly too optimistic, but otherwise effective. I think the paper undersells these a bit, needlessly, in favour of a bit too pessimistic a method, which at face value I only see as being relevant when no information about the shift is available. On the whole, I would prefer a more nuanced exposition of the experiments than the one given.

Besides the above, I only have minor quibbles to do with clarity of notation:

- The expression $U_{CP}(C_\tau, S_m, V, w,b, \delta_C)$ is hard to parse, especially since its actual definition as $U_{CP}( C_\tau, T_N(S_m, V, w,b), \delta_C)$ is hidden away in a statement. For instance, one thing I was missing because of this was that for each possible $w_\tau,$ the set $T_N$ must be resampled. I think this notation should be clarified, perhaps by separating out the definition to draw enough attention to it.

- The expression $s(y|x)$ is unfortunate since $y$ is already used for classes. Perhaps a different letter should be used here.

**Summary Of The Paper:**

This paper is concerned with learning prediction sets (in a PAC sense) under the covariate shift assumption, given a model $f(x,y)$. The form of the sets is restricted to $C_\tau(x) := \{y : f(x,y) \ge \tau\},$ and the problem is set up as learning a $\tau$, using a  labelled source dataset $S_m \sim P^{\otimes n}$ and an unlabelled target dataset $T_n \sim Q^{\otimes n}$ such that with high probability over $S_m, T_n,$ $Q(Y \in C_\tau(X)) \ge 1-\varepsilon,$ where $\varepsilon$ is a given target coverage level. It is desired that $\tau$ is as large as possible to minimise the size of the prediction sets learned.

The main scheme is presented modularly. The paper first describes how finding the maximum $\tau$ whilst ensuring that the number of points in $S_m$ it captures is large enough (as specified by using a Binomial tail inverse) gives valid sets when $Q = P$. Next, it is argued that when $Q$ is absolutely continuous with respect to $P$, and the derivative $w(x) = \frac{\mathrm{d}Q}{\mathrm{d}P}(x)$ is upper bounded and known, then one can importance sample the set $S_m$ to generate a sample from $Q$, which can then plug into the previous procedure. This step fundamentally uses the covariate shift assumption, and, to my understanding, is folklore. The following, then, constitute the main technical contributions.

Next, the assumption of exact knowledge of $w$ is relaxed, and it is argued that if instead for each $x_i \in S_m,$ bounds $\underline{w}_i \le w(x_i) \le \overline{w}_i$ are available, then one can produce a worst-case estimate of the coverage of $C_\tau$ for any $\tau$ (by taking points that were missed to have high weights, and points covered to have low weights), and if this pessimistic coverage also satisfies the constraint. So, for each $\tau$, the worst $w_\tau$ can be produced, which can then feed into the importance sampling procedure above.

Then, it is pointed out that such a confidence bound on the $w_i$s can be learned using a probabilistic classifier $s(\cdot|x)$ to separate data from $S_m$ and $T_n$, which leads to the main proposed algorithm. While it is roughly justified in the appendix that an accurate estimate can be obtained under smoothness assumptions with an appropriately fine gridding of the space (and a consequently huge sample complexity), the concrete proposal is to replace this step with a heuristic method for fitting bounds on the weights, and then plugging these into the above strategy. Note that this is not implemented as a direct optimisation over $\tau$ - instead a set $\mathcal{T}$ of possible values is pre-selected, and the procedure is executed for each $\tau$ (the algorithm recommends an increasing order on the same).

Finally, the paper presents experiments in the DomainNet dataset, and in the ImageNet dataset, where the shift in the latter corresponds to adversarial perturbations. The principal baselines are the weighted split conformal inference (WSCI) method, which in my opinion is an appropriate choice, and the "PS-C" method, which simply uses a prediction set that has $1-\varepsilon/b$ coverage on the source data, where $b$ is an estimated upper bound on $w$. The proposed method is ablatively presented, and it is seen that on just the DomainNet data, the method PS-R (which simply takes $(1-s)/s$ as an estimate of $w$) is both reliable and performs well, while the PS-M method, which further integrates samples in bins tends to be slightly optimistic on this dataset. Conversely, in the adversarially perturbed dataset, PS-R produces trivial prediction sets (since presumably this shift goes entirely outside the source domain's support), while PS-M performs well. The proposed method, PS-W, which further uses upper and lower bounds on $w$s after integrating them over bins, tends to be pessimistic (error-rates of $0.06-0.07$ are observed when only $0.1$ is demanded), but is not too much worse than either of these methods, and is effective in both types of shifts.

**Summary Of The Review:**

I think this is a good paper, which makes an interesting methodological contribution to an increasingly relevant subfield. While it would benefit from better contextualisation of the hyperparameters of the method, and from a softer touch in interpreting experiments, I feel confident in recommending it for acceptance.

---

> ### Author Response · Authors · 2021-11-23
> **Author Response**
>
> Thanks for your valuable comments and suggestions; we have responded below and updated our paper accordingly. We also highlighted the updated parts in red.
>
> ---
> > Firstly, no real justification for the choice of E=0.01 and K=10 is presented. How were these numbers arrived at? Is it reasonable that such crude chunking of the data yields such regular marginals? For that matter, no contextualisation of these numbers is given either - how optimistic or not is assuming E=0.01? Finally, sensitivity with respect to these hyperparameters is not discussed. I think understanding these aspects is critical to the practical usage of a method like this.
>
> *On the choice of $K$.* Our bin is defined in one dimensional space (as mentioned in Appendix B.2), so we follow the usual practice in the calibration literature with respect to binning (e.g., Guo et al., 2017 or Park et al., 2021), where $K$ is about 10 to 20. As we are using equal-mass binning, we choose the number of bins so each bin contains sufficiently many source samples (e.g., 5000 samples) for the Clopper-Pearson interval over IWs for each bin is small (e.g., around 1e-3 for various confidence levels); this leads us to $K=10$. We also added a sensitivity analysis for $K$ for all shifts to Appendix G.6.
>
> *On the choice of $E$.* We use $E=0.001$ in all experiments; we updated this typo. We believe that estimating $E$ (i.e., computing the integral of the difference of source and target probabilities) is intractable since it requires us to perform density estimation in a high-dimensional space (i.e., 2048 in our case), and then compute an integral over each bin. In practice, we find that we can simply choose $E=0$ (or close to zero, e.g., 0.001). Intuitively, we find that this choice works well since binning by the source-discriminator score $g(x)$ is an effective way to group points with similar IWs; rescaling the IWs is only necessary since $g(x)$ might be overconfident, leading to inflated IWs. In Appendix G.5, we provide a sensitivity analysis for E for all of our shifts. Importantly, note that PS-W with E=0 satisfies PAC criterion in all cases.
>
> We have added our discussion to Appendix C.3
>
> ---
> > Secondly, I don't think the conclusion is as clear as is suggested by the "mean-normalised-size-of-reliable-methods" metric. This mean normalised size itself is not too meaningful as a metric to me (these are fairly different tasks with fairly different scales of difficulty as evinced by the mean sizes themselves). More importantly, I find it a much more natural statement to say that PS-R as a method makes sense when the shift is non-adversarial, and PS-M as a method that is slightly too optimistic, but otherwise effective. I think the paper undersells these a bit, needlessly, in favour of a bit too pessimistic a method, which at face value I only see as being relevant when no information about the shift is available. On the whole, I would prefer a more nuanced exposition of the experiments than the one given.
>
> Thanks for the suggestion; we have added a more nuanced exposition of the experiment's interpretation in Section 4.2.
>
> ---
> > The expression $U_{CP}(C_\tau, S_m, V, w, b, \delta_C)$ is hard to parse, especially since its actual definition as $U_{CP}(C_\tau, T_N(S_m, V, w, b), \delta_C)$ is hidden away in a statement. For instance, one thing I was missing because of this was that for each possible $w_\tau$ the set $T_N$ must be resampled. I think this notation should be clarified, perhaps by separating out the definition to draw enough attention to it.
>
> We have highlighted the definition in Equation (5) to draw attention.
>
> ---
> > The expression $s(y|x)$ is unfortunate since $y$ is already used for classes. Perhaps a different letter should be used here.
>
> We now use $s$ instead of $y$ and use $g$ instead of $s$, i.e., use $g(s|x)$ instead of $s(y|x)$.

---

### Decision · Program_Chairs · 2022-01-20

**Decision:**

Accept (Poster)

**Comment:**

All reviewers were clear in their opinion that the paper deserves to be accepted. One reviewer also indicated a wish to increase the score from 6 to 7 but was not able to do that, so it isn't reflected in the final score. The reviewers appreciated the methodological contribution made by the paper.